# Effectiveness of a chatbot in improving the mental wellbeing of health workers in Malawi during the COVID-19 pandemic: A randomized, controlled trial

Eckhard Kleinau[1]☯*, Tilinao Lamba[2]☯, Wanda Jaskiewicz[3], Katy Gorentz[3], Ines Hungerbuehler[4], Donya Rahimi[3], Demoubly Kokota[2], Limbika Maliwichi[2], Edister Jamu[2], Alex Zumazuma[5], Mariana Negrão[4], Raphael Mota[4], Yasmine Khouri[4], Michael Kapps[4]

1 University Research Co. (URC), Chevy Chase, Maryland, United States of America, 2 Dept. of Psychology, University of Malawi–Chancellor College, Zomba, Malawi, 3 Global Health Division, Chemonics International, Washington, District of Columbia, United States of America, 4 Clinical Division, Vitalk, São Paulo, Brazil, 5 Department of Mental Health, Kamuzu University of Health Sciences (KUHES), Blantyre, Malawi

☯ These authors contributed equally to this work.
* ekleinau@urc-chs.com

**Data Availability Statement:** Deidentified study data are provided as supplemental information.

## Abstract

We conducted a randomized, controlled trial (RCT) to investigate our hypothesis that the inter-active chatbot, Vitalk, is more effective in improving mental wellbeing and resilience outcomes of health workers in Malawi than the passive use of Internet resources. For our 2-arm, 8-week, parallel RCT (ISRCTN Registry: trial ID ISRCTN16378480), we recruited participants from 8 professional cadres from public and private healthcare facilities. The treatment arm used Vitalk; the control arm received links to Internet resources. The research team was blinded to the assignment. Of 1,584 participants randomly assigned to the treatment and control arms, 215 participants in the treatment and 296 in the control group completed baseline and endline anxiety assessments. Six assessments provided outcome measures for: anxiety (GAD-7); depression (PHQ-9); burnout (OLBI); loneliness (ULCA); resilience (RS-14); and resilience-building activities. We analyzed effectiveness using mixed-effects linear models, effect size estimates, and reliable change in risk levels. Results support our hypothesis. Difference-in-differences estimators showed that Vitalk reduced: depression (-0.68 [95% CI -1.15 to -0.21]); anxiety (-0.44 [95% CI -0.88 to 0.01]); and burnout (-0.58 [95% CI -1.32 to 0.15]). Changes in resilience (1.47 [95% CI 0.05 to 2.88]) and resilience-building activities (1.22 [95% CI 0.56 to 1.87]) were significantly greater in the treatment group. Our RCT produced a medium effect size for the treatment and a small effect size for the control group. This is the first RCT of a mental health app for healthcare workers during the COVID-19 pandemic in Southern Africa combining multiple mental wellbeing outcomes and measuring resilience and resilience-building activities. A substantial number of participants could have benefited from mental health support (1 in 8 reported anxiety and depression; 3 in 4 suffered burnout; and 1 in 4 had low resilience). Such help is not readily available in Malawi. Vitalk has the potential to fill this gap.

**Funding:** WJ Financial support for this RCT was provided by the United States Agency for International Development under the Cooperative Agreement No. AID-OAA-A-15-00046 https://www.usaid.gov The funders had no role in study design, data collection and analysis, decision to publish, or preparation of the manuscript.

**Competing interests:** The authors have declared that no competing interests exist.

## Introduction

Healthcare provision can be stressful even under normal circumstances, and maintaining the mental wellbeing of health workers is of utmost importance for optimal and safe patient care [1]. The COVID-19 pandemic has overwhelmed countries' health systems and increased care-related pressure to ensure patient and staff safety. Health workers confront life and death decisions, physical exhaustion, lack of protective equipment, and fear of infection as daily threats to their mental wellbeing. An increasing number of studies report a high proportion of health professionals globally suffering from depression, anxiety, and burnout. A systematic review of 59 studies found that a median of 24% of health workers suffered from anxiety, 21% from depression, and 37% from distress [2]. A meta-analysis from 36 countries reported slightly higher levels (8.0% for depression; 26.9% for anxiety; 24.1% for post-traumatic stress symptoms; 36.5% for stress; and 50.0% for distress) [3]. Recent studies from sub-Saharan Africa showed similar levels. A multi-center cross-sectional study from Ghana, found the following levels: depression (21.1%), anxiety (27.8%) and stress (8.2%) [4]. Research from Ethiopia found the following prevalence: depression (20.2%), anxiety (21.9%), and psychological distress (15.5%) [5]. Data from South Africa suggest high levels of mental disorders (around 50%) among all types of health workers due to work-related stress [6]. A study from Malawi using a small sample of nurses and the Coronavirus Anxiety Scale suggests that 26% (n = 26) of respondents had COVID-19-related anxiety and 48% (n = 49) functional impairment [7].

Mental disorders not only pose threats to patient safety and health workers' quality of life, but they also incur high economic costs. A World Health Organization (WHO)-led study, well before the COVID-19 pandemic, found that depression and anxiety disorders cost the global economy US$1 trillion each year [8]. This study estimated a return on investments of between 2.3 and 5.7 to 1 if treatment of these mental disorders was scaled up. Mental health interventions include basic psychosocial counselling for milder cases, and more intensive psychosocial treatment plus antidepressant drugs for more severe cases. Over the last decade, computer- or Internet-based cognitive behavioural therapy (c-CBT or i-CBT) has been tested and implemented as alternatives to in-person treatment. Research has shown that c-CBT-based self-administered interventions improve depression and anxiety in adults. A meta-analysis of 49 randomized controlled trials (RCT) revealed a significant medium to large effect size (Hedges' g = 0.77, 95% CI 0.59 to 0.95) of c-CBT for depression and anxiety [9].

Another meta-analysis of 22 RCTs found an even greater effect size (g = 0.88, 95% CI 0.76 to 0.99) [10]. However, a recent systematic review found small to medium post-treatment pooled effect sizes regarding depressive symptoms (g = 0.51, 95% CI 0.30 to 0.72) and anxiety symptoms (g = 0.44, 95% CI 0.23 to 0.65) of c-CBT for reducing these symptoms in adolescents and young adults compared to passive controls [11]. Clinical trials have established that mobile applications can effectively deliver CBT programs for the treatment of: depression [12]; self-management of chronic pain conditions [13]; and social anxiety disorder [14]. C-CBT or i-CBT can lower barriers to seek help, which is especially important in settings such as those in sub-Saharan Africa with low levels of access to psychologists and therapists [15].

C-CBT and i-CBT have evolved into interactive, automated conversational agents (chatbots) that are driven by artificial intelligence, including the recent entrant Vitalk (with versions available for the public and adapted for health professionals). A panel study without control group in Brazil established the effectiveness of the public version, finding a large post-intervention effect size of Cohen's d of -0.81 or greater for anxiety, depression, and stress [16]. The version for health workers has been pilot tested in Malawi, a country in which access to mental health therapy is very limited, with only 0.02 psychologists, 0.01 psychiatrists and 0.04 occupational therapists per 100,000 population [17]. Only recently has the country successfully

trained 3 psychiatrists. The frequent deployment of mental health professionals, especially psychiatric nurses, to other duties such as maternity services exacerbates this situation [18].

To date, there have been few studies, and even fewer RCTs, with the aim of establishing the effectiveness of chatbots. Most of these trials were based on small samples of 70 participants or fewer, had a short duration of 2–4 weeks, suffered from serious biases, and fell short of establishing that chatbots lead to improved mental health outcomes. All trials were conducted in high-income countries, and none in low-resource settings [19]. No RCTs have been conducted to assess the effectiveness of chatbots in improving the mental health status of health workers specifically [15].

To fill these gaps in evidence, we conducted an RCT in Malawi (building on the earlier pilot test), to investigate our working hypothesis that a virtual mental healthcare assistant chatbot, Vitalk, is an acceptable source of psychosocial and mental wellbeing support for health workers to effectively decrease work-related anxiety, depression, burnout, and loneliness (based on standard mental health scales), and to increase resilience and resilience-building behaviours.

## Methods

### Study design

This study, a 2-arm, parallel RCT (S1 File), with a pre-treatment assessment, 8-week intervention period, mid-study assessments at 4–5 weeks, and an end-of-study assessment at 9 weeks, was conducted by 2 study teams with distinct roles: a trial management team and a research team. The management team reviewed registration data, removed exclusions, assigned groups and unique trial IDs, and provided communications and technical support to participants. The research team developed the study protocol, recruited participants, held participant workshops at the beginning and end of the study, held focus group discussions (FGDs) during the final workshop, and analyzed the data (de-identified before the analyses).

Participants were randomly assigned in equal proportions to either the treatment or control group after registration. The trial management team did the random assignment using permuted block randomization sequences but concealed the allocation and identifying information from the research team for the duration of the study. Vitalk, the developer of the app, created a web portal for participants to register, give consent, and enter their demographic information online. Participants received an email with their unique trial ID and a link to either the treatment app, Vitalk, or to the website (control group). As a single-blinded design, only the research team was blinded to the study arm assignments; participants in both arms were told they were participating in a study on "online self-help for mental wellbeing" but they were aware of the specific intervention they were using and may have found out that others used a different intervention. Neither personally identifiable information nor the participants' facility affiliations were shared outside the trial management team.

### Study population and recruitment

The 8 professional cadres (Table 1) eligible for participation were recruited from all public and private primary, secondary and tertiary care facilities within Blantyre and Lilongwe districts.

Table 1. Type of health workers eligible.

| | |
|---|---|
| • Doctors | • Nurses |
| • Medical Assistants | • Clinical officers |
| • Laboratory technicians | • Physiotherapy technicians |
| • Pharmacists | • Physiotherapists |

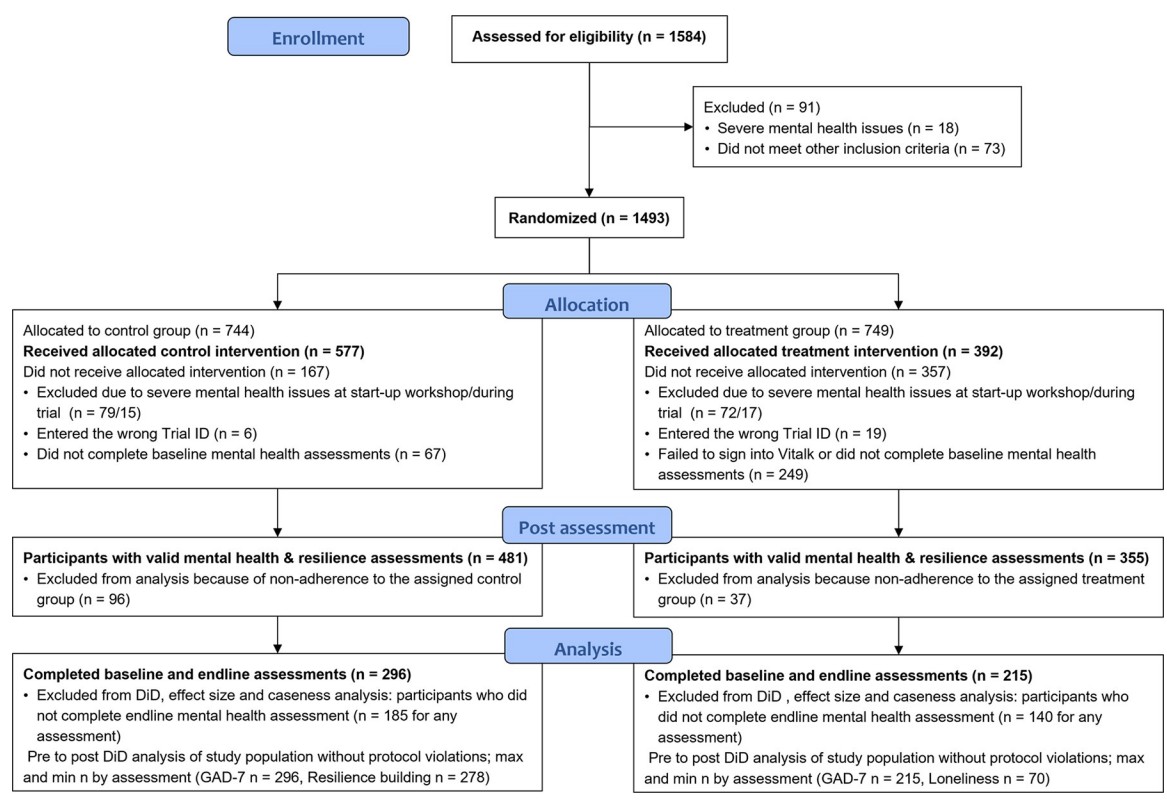

**Fig 1. Study participant flow.**

We recruited participants via posters placed in participating health facilities. All facilities were contacted by phone and email; in larger facilities the research team made short presentations to leadership to seek permission to recruit from their facilities and hang the posters.

The Consolidated Standards of Reporting Trials (CONSORT) diagram in Fig 1 shows that of 1,584 participants enrolled in the study, 836 completed one or more mental health and resilience assessments (481 in the control and 355 in the treatment group). Based on participant feedback, the treatment group saw a much greater attrition due to technical difficulties impacting the use of the Vitalk app.

## Informed consent, screening, and eligibility

Interested health workers were invited to a half-day workshop to provide general information and answer questions about the study, and in which they could choose to join the study by signing an informed consent form (kept under lock by the University of Malawi). The research team introduced the trial as a "mobile-phone-based research study of online self-help for mental wellbeing" but provided no details about the 2 study groups.

To be eligible, participants needed to meet the following inclusion criteria:

- Be currently employed as 1 of the 8 types of service providers listed above (a minimum educational qualification of a diploma adhering to the Ministry of Health employment criteria; medical assistants with a certificate and 2 years of college completed qualified)

- Possess some degree of English language proficiency

- Own a smartphone with an Android operating system

- Have no history–past or current–of counseling or therapy for severe mental health disorders

- Not have self-reported suicidal ideation (question 9 of the PHQ-9)

- Score below "very high risk" levels for depression, or below "very high risk" of anxiety combined with "high" or "very high risk" of depression on the initial and subsequent assessments

The management team emailed excluded participants Internet resources for self-help. For those excluded based on suicidal ideation or very high risk of depression and/or anxiety, the team also sent the contacts of 2 local psychologists.

## Study ethics approval, procedures, and participant retention

The University Research Co. (URC, no reference #) Institutional Review Board and the University of Malawi Research Ethics Committee (UNIMAREC), approved the study protocol (reference # P.09/20/84) on October 8 and 15, 2021, respectively (S2 File). The trial was registered retrospectively with the ISRCTN registry (trial registration number: ISRCTN16378480), because the registration process took longer than anticipated and was delayed due to administrative constraints. The authors confirm that all ongoing and related trials for this intervention were registered.

In Malawi, Internet access and the use of apps through mobile phones requires expensive data packages. Participants received 3 data packages of 5 gigabytes (GB) valued at US$10 each (distributed during the startup workshop, halfway through the study, and at the final workshop). This data allowance covered a minimum of at least 1 month of an hour of daily use of Vitalk or web resources. At the startup and final workshops, participants received a transport allowance (US$10) and a food allowance (US$5), following government guidelines. These allowances were not tied to active participation in the trial to avoid any undue influence, and participants received no additional incentives. As the principal means of participant retention and optimizing the response rate, both study groups received weekly email and WhatsApp messages encouraging them to engage with their app or website resources. During the baseline period, the management team provided the treatment group with detailed download and sign-in instructions for the app via WhatsApp, after noticing low sign-in numbers and receiving messages about sign-in difficulties. The research team was blinded to this process.

## Study timeline

Participant recruitment took place between October 15 and 24, 2021. Ethics approval and participant recruitment took longer than planned in the protocol, which delayed the start of the study by 1 week. The study spanned 56 days (October 25-December 19, 2021), with startup workshops the week of October 18 and end-of-study workshops the week of December 20 (Fig 2). The research team held several onboarding workshops each time to accommodate the large number of participants and their varied work schedules, while complying with physical distancing requirements due to COVID-19. The length of the study enabled treatment group participants to complete exercises for 2 "carelines" (thematic conversations addressing anxiety, depression, stress, relationships, or the COVID-19 pandemic) over a period of a month for each. The baseline, midline, and endline periods take into account that participants completed their first mental health assessments at varying times after registration and repeated assessments with varying frequencies before completing their selected carelines. Several participants completed assessments in a post-endline period after the study concluded; they were excluded from the analysis.

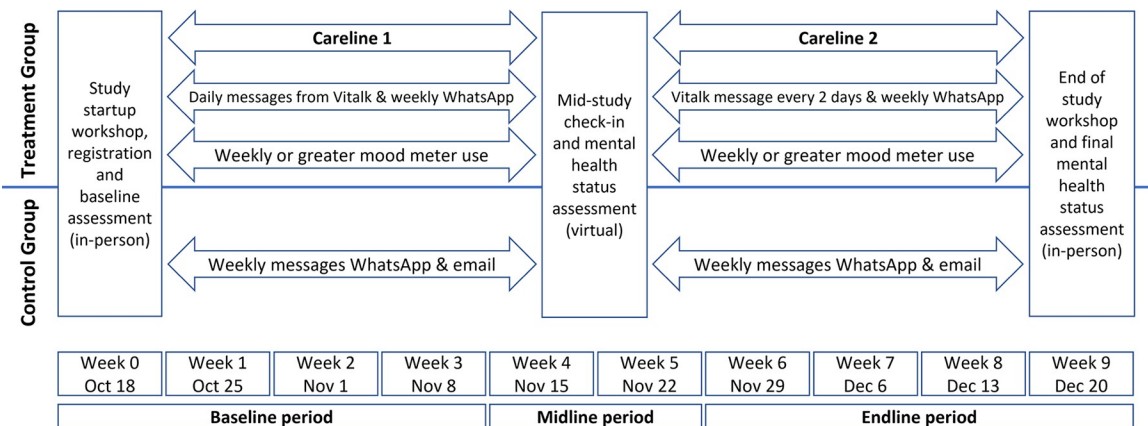

**Fig 2. 2021 Study timeline.**

## Sample size and randomization

The required study population size for paired tests of correlated means was calculated using STATA 17 and is based on the following information from published literature:

- Pre-post-treatment difference of 1.5 points on standard scales for mental health assessments using 2-tailed tests

- Standard deviation of the difference = 6

- Power 80%

- Significance level 5%

This resulted in a minimum of 128 participants per study arm. Given the continuity experience (the proportion of participants completing the pre- and post-treatment assessments for depression, anxiety and stress) of 20–45% reported by Daley et al. [16], we assumed that the dropout rate could be as high as 75% for at least 1 of the standard mental health assessments. This required a sample of 512 participants per arm to yield an effective post-intervention sample of 128. Furthermore, if 20% of potential participants did not meet inclusion criteria or dropped out for other reasons, about 640 enrollees were needed initially per arm.

The management team assigned randomized trial participant numbers (trial ID) using permuted block randomization sequences, generated by the website Randomization.com [20], to maintain a balance across the study arms. Block size varied between 4 and 10 randomly ordered treatment assignments. A random sequence of trial IDs between 1001 and 3000 was generated using the Research Randomizer [21]. We achieved allocation concealment by charging the trial management team with group assignments using randomized trial IDs and permuted block randomization. Assignment and personal identifying information was not shared with the research team responsible for data analysis to ensure that it remained blinded throughout trial implementation.

## Study interventions

**Treatment group.** Participants in the treatment group were asked to download and register with the mental health chatbot app, Vitalk, and to use it daily over the 8-week study period. The aim of Vitalk is to improve mental wellbeing by reducing stress, anxiety, depression, and burnout and increasing resilience using a preventative approach to mental health. Vitalk is an

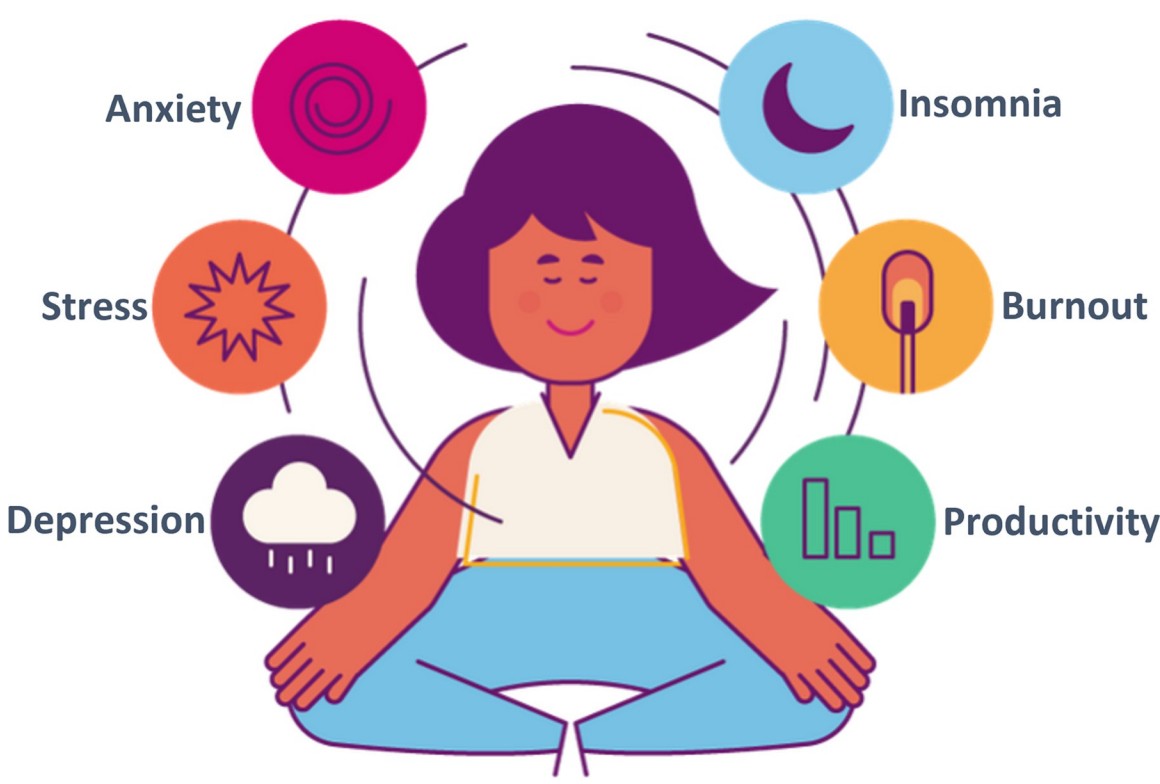

**Fig 3. Viki, Vitalk's virtual mental healthcare assistant.**

automated chatbot, using an avatar named Viki (Fig 3) to deliver mental health content in an innovative conversational format.

The conversations are based on insights and strategies taken from Cognitive Behavioral Therapy (CBT) and Positive Psychology [22, 23], which include (but are not limited to): psychoeducation; cognitive restructuring; behavioral activation; gratitude; and practical exercises (breathing, relaxation, and meditation). The goal of the conversations is to help users reflect on experiences and learn techniques to manage stress, mood, and anxiety. A mood tracking tool aids reflection, "emojis" mimic a natural human interaction, and gamification increases user engagement [24]. Vitalk users choose between different thematic conversations ("carelines") and use standardized instruments to assess their mental health and resilience status before and after completing a careline at monthly or more frequent intervals. The app provides users with feedback and advice about their assessment scores. Carelines are continuously refined to ensure the language and emotional expression matches the needs of different users. If user input in the PHQ-9 questionnaire suggests suicidality (e.g., "life isn't worth living") or suicidal ideation, the risk module is triggered and the chatbot delivers information about emergency support, including the contact information of local psychologists.

The version used in Malawi consisted of 5 carelines: anxiety; depression; stress; relationships; and the COVID-19 pandemic. Throughout the 56-day study period, users selected 2 carelines to complete (4 weeks for each). During the first careline period (the first 4 weeks of the study), the user was engaged in daily conversations, each lasting 5–10 min. This changed to an engagement every other day during the second careline period (4 weeks). After the study, users were free to continue using the app and experience the other carelines, receiving a final data bundle at the last workshop to enable them to do so.

The chatbot avatar Viki was developed through focus groups and pilot testing in Brazil. A team of clinical psychologists and healthcare professionals from Vitalk wrote the conversations used by Viki and adapted them to the Malawi context. Unlike that of Brazil, the Malawi version did not include natural language understanding (NLU). The chatbot was styled and designed by an information technology, product development and user experience (UX) team at Vitalk in São Paulo. The app is built on ruby and JavaScript as a free-to-use service, hosted within an instant messenger platform, accessible from Android devices.

Vitalk does not aim to replace a healthcare professional or to offer treatment. Users are made aware of this limitation in the terms and conditions required for consent; they are advised to seek additional support if they show a high risk of depression, anxiety, or burnout in their mental health assessments.

**Control group (waitlist).** Participants in the control group were provided access to a webpage in their Internet browser with links to 4 mental health resources: Mental Health Foundation in the United Kingdom [25]; WHO/Geneva mental well-being resources for the public [26]; WHO/Geneva #HealthyAtHome—Mental health [27]; and the Doing What Matters in Times of Stress: An illustrated Guide [28]. These resources had no interactive features (passive) and relied on participants' initiative to access them. Control group participants were considered to be wait-listed since they were given access to the Vitalk app after the completion of the study.

## Outcomes

**Primary outcome measures.** This study used the scores from several standardized mental health and resilience assessments as outcome measures (S3 File). In addition, we introduced a new measure of resilience-building activities. In the analysis, we used total assessment scores as well as risk groupings to establish clinical relevance following classifications from published literature (based on the sensitivity and specificity of the assessments). All mental health and resilience questionnaires were pilot tested by Vitalk and members of the research team in Malawi.

*Patient Health Questionnaire (PHQ-9)*. The PHQ-9 is a 9-item self-reported scale that evaluates symptoms of depression over the past 2 weeks (e.g., "how often have you been bothered by feeling down, depressed, or hopeless"). Item response options use a Likert scale ranging from 0 (not at all) to 3 (nearly every day), with a maximum score of 27. According to the literature, total scores are divided into 5 risk categories: none (0–4); mild (5–9); moderate (10–14); moderately severe (15–19); and severe (20+) symptoms. The PHQ-9 has been widely used and validated in Malawi [29].

*Oldenburg Burnout Inventory (OLBI)*. Burnout is linked to relatively high work requirements and limited resource availability for managing them. The discrepancy between resources and challenges creates a significant negative emotional state. The OLBI has 16 items rated 1 to 4, with a maximum score of 64. Eight items describe exhaustion (OLBI-E) and 8 describe disengagement (OLBI-D) [30]. The questionnaire includes both straight and reversely worded items in both dimensions. According to the literature, total scores are divided into 3 risk categories: low (16–35); moderate (36–43); and high (44+). OLBI has been validated and used in low- and middle-income countries (LMICs) [31].

*Generalized Anxiety Disorder (GAD-7)*. The GAD-7 is a 7-item self-reported scale to assess anxiety symptoms over the past 2 weeks (e.g., "how often have you been bothered by feeling afraid something awful might happen"). Scores range from 0 (not at all) to 3 (nearly every day), with a maximum score of 21. The total scores are divided into 4 risk categories: none (0–4); mild (5–9); moderate (10–14); and severe (15+) symptoms. GAD-7 has been used effectively in Malawi and other LMICs with comparable demographics [32].

*UCLA short (three-item) Loneliness scale (UCLA Loneliness)*. UCLA Loneliness was added to Vitalk for this study. While there is no agreed upon definition of "loneliness" in research, it is described as an unwelcome, painful, and unpleasant feeling, and a fluid experience that can come and go over a short time or persist in the longer term. The UCLA Loneliness scale consists of 3 items rated 1 to 3, with a maximum score of 9. Total scores are divided into 2 risk categories: non-lonely (3–5) and lonely (6+) [33, 34].

*5-item resilience-building behavior scale*. A novel addition to this study and Vitalk was an assessment of participants' resilience-building behaviors related to stress management, self-awareness, self-care, purpose, and connection with others (adapted from a toolkit for building health worker resilience [35]). While numerous scales for measuring resilience exist, we did not find any publications on the validity and reliability of instruments measuring resilience-building activities. The questionnaire had 5 items rated 1 to 4, with a maximum score of 20, related to the frequency with which participants performed resilience-building activities over the past 2 weeks. Scores were divided into 3 activity levels: low or none (5–10); moderate (11–15); and high (16–20).

*14-item Resilience Scale (RS–14)*. Resilience refers to the ability to withstand or adaptively recover from stressors, promoting psychological and physical wellbeing. Resilience is negatively correlated with symptoms of generalized anxiety and post-traumatic stress and positively correlated with gratitude, optimism, and positive affect. The 5 characteristics of resilience are: meaningful and purposeful life; perseverance; equanimity; self-reliance; and existential aloneness [36]. The RS-14 has 14 items rated 1 to 7, with a maximum score of 98. Total scores are divided into 6 resilience levels: very low (14–56); low (57–64); on the low end (65–73); moderate (74–81); moderately high (82–90); and high (91+). RS-14 has been validated and used in LMICs [37, 38].

While the app registers the actual scores, to simplify communication of risk to users, Vitalk reports 4 levels of mental health risks and resilience for its 4 standard assessments (GAD-7, PHQ-9, OLBI, and RS-14): very low; low; high; and very high risk. The Vitalk risk classification differs from published levels as follows: 1 additional level for OLBI; 4 instead of 5 levels for PHQ-9; and 4 instead of 6 levels for RS-14. For the purposes of this trial, we used only published risk classifications based on actual test scores.

Participants completed all 6 assessments online, which together required about 20 minutes. Participants in both arms were allowed to take the assessments repeatedly and saw their results immediately after answering the 6 questionnaires. Participants were prompted either by the avatar Viki or What's App messages and emails to take the assessments at specific intervals during the baseline, midline, and endline periods.

**Secondary measures.** *Mood meter*. Users of the Vitalk app had the option of assessing their mood using emojis representing 5 mood levels (0–4): very bad; bad; okay; good; and very good. Emojis are successfully used in sentiment analysis and show a stronger emotion compared to words [24]. The timing of mood assessments was matched to mental health assessments prior to data analysis since participants took them independently. The mood meter was part of the intervention to raise self-awareness, and only available to the treatment group.

*Participant engagement*. For the treatment group, we captured several measures related to participant engagement over the study period: number of interactions with Vitalk; number of days with interactions with the app; number of days between the first and last day of interaction with the app; and total number of hours spent on the app. For the control group, we captured 2 measures related to participant engagement: number of days they accessed web resources and number of web resource click-throughs. All raw engagement measures were converted to z-scores for both groups to account for differences in metrics and distributions in each group. Standardized hours spent on Vitalk, and standardized number of days with web resources accessed, were retained as determinants of mental health and resilience scores.

*Smartphone experience.* During registration, we measured participants' smartphone experience by asking how often (never, less often, weekly, several times a week, daily) they used the following features (each scored 0 to 4): Internet; WhatsApp; Facebook, Instagram, or twitter; chat app; mobile money; games; email; videos; and music. Smartphone experience is a composite of the frequency of use for all 9 features, with a maximum score of 36.

*Current or past exposure to mental health counseling.* During registration, participants were asked whether they had any current or past exposure to mental health counseling. Possible responses were currently receiving counseling; received counseling in the last 6 months; received counseling in the last 2 years; received counseling more than 2 years ago; and never received counseling. Participants with any counseling experience were then asked whether counseling was for mild or moderate mental health issues, or for acute or severe mental health issues. Participants who responded to the latter were excluded from the study and given the contact for local psychologists.

*Current or past chat app and website use for mental health.* At registration, participants were asked about their current use of a chat app for mental wellbeing (with the response options: never; sometimes; and often). Past use was assessed through a binary yes or no response. We created a composite "chat app for mental health" variable with values ranging from 0 (no chat app use) to 5 (current often use and past use) by combining responses from both questions. Current and prior use of Internet-based resources about mental wellbeing was assessed in the same manner.

*Impact of COVID-19.* Four questions assessed how the COVID-19 pandemic impacted study participants. The first 3 questions asked were about the impact on workload, workhours, and stress levels over the past 12 months. Possible response options for each question (scored from 0 to 4) were: greatly decreased; somewhat decreased; stayed the same; somewhat increased; and greatly increased. The sum of the responses to these questions created a composite COVID-19 effect variable. The fourth question inquired whether COVID-19 prevented participants from going into work over the past 12 months. Response options (scored from 0 to 3) were: no disruptions at all; several days over the entire year; several days every month; and several days every week. Participants who did not work at all during this period were coded as missing.

Participants completed an online questionnaire about their smartphone experience, current or past exposure to mental health counseling, current or past chat app and website use for mental health, and impact of COVID-19.

*Anonymous online participant experience questionnaire.* During the final workshops, participants from both arms had the opportunity to complete an anonymous online questionnaire about their experience using either the Vitalk app or the web resources. Questions focused on the frequency of use, reasons for infrequent use, relevance and usefulness, ease of use, and specific difficulties encountered. Some key findings related to study outcomes are highlighted in this paper; a separate comprehensive analysis will be published together with the findings from focus group discussions.

## Data analysis

Participant characteristics are presented as proportions or means (SD) for the treatment and control groups.

**Mixed linear model.** We used multilevel mixed-effects generalized linear models (STATA *meglm* command) to analyze primary participant-reported outcome measures for all 6 mental health and resilience assessments over the baseline and endline periods. A full factorial model of study group and assessment period was fitted to represent fixed effects. Participants' trial ID

specified the random effect in this model. The parameter estimate of interest was the coefficient of the interaction between study group and assessment period, which is equivalent to a difference-in-differences (DiD) estimator. The DID estimator compares the differences in outcomes before and after treatment for the treatment group with any changes in the control group. It is a measure of additional change between base- and endline in the treatment group, if the value is statistically significant at a 0.05 level or smaller. DiD estimators were robust, because only valid assessments for the baseline and endline periods were included in the analysis, avoiding the need for inter- or extrapolating missing values. Because of randomization, covariates were not included in the mixed models. Data from the midline and any assessments completed after the 8-week study period (post-endline) were not included in the analysis because of a much lower number of observations in these compared to the other periods.

**Effect size.** To compare the different outcome measures, Cohen's d was calculated as the mean difference between the 2 groups divided by their common SD and interpreted as small (d = 0.2), medium (d = 0.5), and large (d = 0.8).

**Reliable change index.** To assess whether the change in outcomes (the total score for each assessment, between baseline and endline) is statistically reliable, and not due to measurement variability inherent to the assessments alone, we calculated a reliable change index (RCI). The RCI is the standard error of the difference between the 2 measurements, $SE_{diff}$, in Eq (1). $SD_{bas}$ is the standard deviation of the baseline observations, and r is Cronbach's alpha representing the reliability of the measurement instrument. Each assessment instrument has its own RCI.

$$RCI = 1.96 \times SE_{diff} = 1.96 \times SD_{bas}\sqrt{2}\sqrt{1-r} \tag{1}$$

**Change in risk levels.** We used total scores from the 4 mental health and 2 resilience assessments for the mixed linear model, effect size, and RCI estimation. To determine whether the changes between the pre- and post-intervention periods matter from a clinical perspective, the total assessment scores–a continuous variable–were converted to mental health risk and resilience levels–an ordinal variable–using the published classifications described above for each type of assessment (e.g., depression scores from the PHQ-9 are divided into 5 levels ranging from none to severe). A change in risk levels was calculated as either a drop or increase of 1 or more levels for each mental health and resilience metric. For participants registering a change in risk level, the proportion with reliable change was calculated. Reliable change was established if the difference between baseline and endline scores were equal or greater than the RCI.

**Determinants of mental health and resilience.** All mental health and resilience outcomes were regressed against all participant characteristics using ordinary least squares models, controlling for study group and assessment period. All observations with valid baseline and endline assessments for the control and treatment groups were included in the analysis. A separate analysis was done for the treatment group to assess which covariates unique to this group (mood meter scores and carelines) affected the outcomes.

In general, we chose a p-value of 0.05 or less as the level of statistical significance. However, we also considered larger p-values of p ≤ 0.10 as trending towards statistical significance, which may be clinically relevant for improving the practice of self-help. Equally important as the p-value was the consistency of patterns across all mental health and resilience assessments and their agreement with behavioral theory. All data (S4 File) were analyzed with STATA 17.0 (StataCorp LLC, College Station, TX).

## Results

Out of 1493 study participants randomized 392 received the treatment and 577 were wait listed controls. These numbers dropped to 355 in the treatment arm and 481 in the control arm

**Table 2. Number and percent of participants completing any mental health and resilience assessment by study period.**

| Study period | Control Group n | Control Group % | Treatment Group n | Treatment Group % | Total n |
|---|---|---|---|---|---|
| Baseline | 469 | 60% | 318 | 40% | 787 |
| Midline | 77 | 30% | 181 | 70% | 258 |
| Endline | 304 | 54% | 260 | 46% | 564 |
| Post-endline | 13 | 13% | 86 | 87% | 99 |

Numbers include singular assessments that were completed in one study period only, which were excluded from analysis. Mid- and post-endline assessment were also excluded from the analysis.

because of non-adherence to study group assignments. A total of 836 participants had valid mental health and resilience assessments. However, the number of participants completing assessments varied substantially by study period, as seen in Table 2.

Using complete case analysis (CCA), only participants who finished base- and endline assessments were included in the calculations of DiD estimators using mixed-effects linear modeling, effect size estimation, and change in risk levels. We used CCA because missing data seem to be random as shown in the CONSORT diagram (Fig 1), which includes access to the Vitalk app or website issues, and conditionally independent of the study outcome measures. Moreover, study population characteristics of all participants with valid mental health and resilience assessments were very similar in the control and treatment groups (Fig 4). The relatively small numbers at midline and post-endline in the control group are possibly related to the passive nature of the intervention for this group. The interactive Vitalk app seems to have enticed assessment behavior in the treatment group, which experienced a much smaller drop.

Table 3 shows the effective sample size and missing data by type of assessment. The much higher proportion of missing data for the loneliness and resilience building assessments is because participants had difficulties accessing these assessments, which were not part of the original Vitalk app but were added for this study. These 2 assessments were the last in the series of the 6 assessments suggesting that a majority of Vitalk users stopped completing assessments after RS-14. The control group did not experience a similar drop. Overall, 511

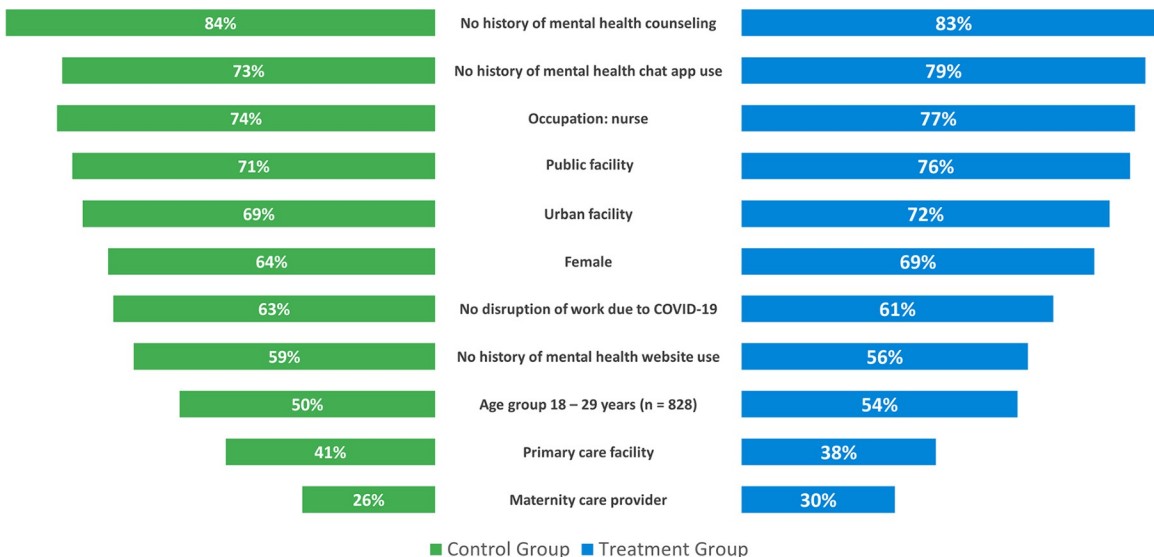

**Fig 4. Study participant characteristics.** Only the most frequent categories are shown (n = 836 [481 control group, 355 treatment group] for all variables except Age with n = 828).

**Table 3. Effective sample size and missing data by type of mental health and resilience assessment with completed baseline and endline.**

| Assessment | Control Group n | Treatment Group n | Total n | Control Missing (%) | Treatment Missing (%) |
|---|---|---|---|---|---|
| Anxiety (GAD-7) | 296 | 215 | 511 | 185 (38%) | 140 (39%) |
| Depression (PHQ-9) | 286 | 201 | 487 | 195 (41%) | 154 (43%) |
| Burnout (OLBI) | 291 | 211 | 502 | 190 (40%) | 144 (41%) |
| Resilience (RS-14) | 280 | 183 | 463 | 201 (42%) | 172 (48%) |
| Loneliness (UCLA) | 280 | 70 | 350 | 201 (42%) | 285 (80%) |
| Resilience building activities | 278 | 85 | 363 | 203 (42%) | 270 (76%) |

Assessments shown in the order completed by participants.

These are the sample sizes used in DiD, effect size, and change in risk level analyses.

participants (296 in the control and 215 in the treatment group) had no protocol violations and completed baseline and endline assessments for GAD-7.

## Participant characteristics

The distribution of characteristics for the 836 participants with valid mental health and resilience assessments was balanced between the control and treatment groups as shown in Fig 4. The difference between groups varied by 1 to 6 percentage points. Overall, participants were young (under 30 years) and female. Most were professional nurses working in public secondary or tertiary healthcare facilities in urban areas. Maternity care was the most common type of care provided. Fewer than 30% had ever used a chat app for mental health and more than 40% had ever accessed websites with mental health content. One in 6 participants had a history of mental health counseling. The level of experience in performing 9 different tasks with their smartphones was high (a mean of around 29 out of a maximum score of 36) and nearly the same in both study groups (Fig 5), corresponding to several times per week to daily use. Both

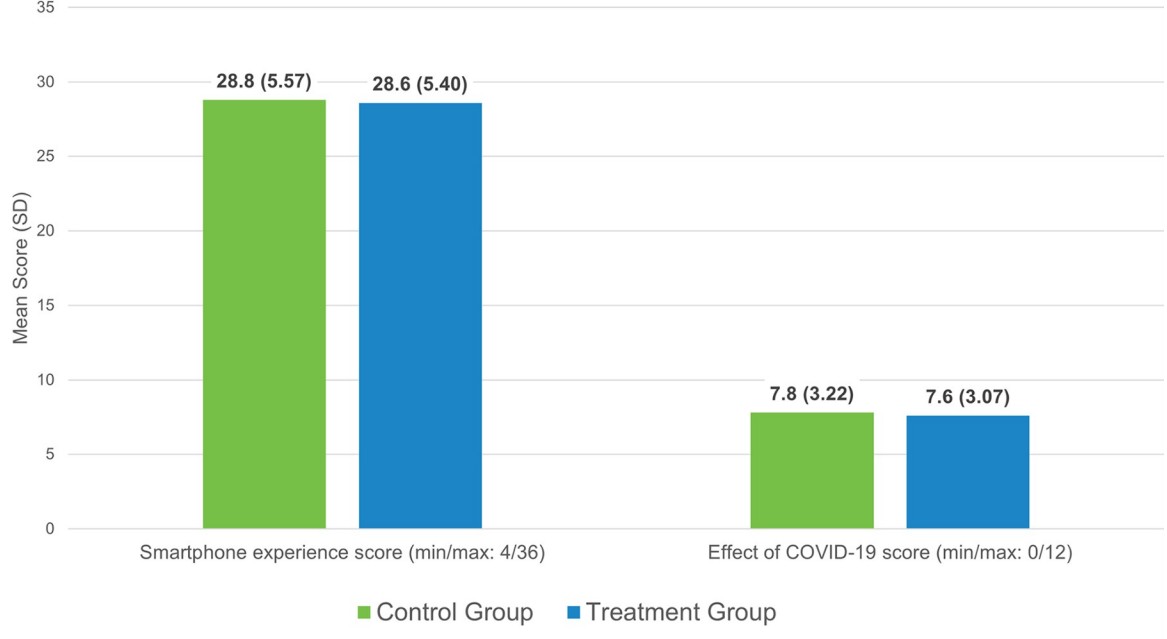

**Fig 5. Mean smartphone experience and COVID-19 impact scores (n = 836).**

**Table 4. Participant adherence to treatment and control interventions.**

| Group/Variable | Mean | Standard Deviation |
|---|---|---|
| **Treatment Group** | | |
| Number of interactions with Vitalk app | 2,200 | 6,294 |
| Duration of app use in days | 24 | 14 |
| Time spent in the Vitalk app in hours | 10.4 | 7.7 |
| Number of time mood meter used | 13 | 15 |
| Mood meter score on a scale 0 (very bad) to 4 (very good) | 2.9 | 1.0 |
| **Control Group** | | |
| Number of days web resources were accessed | 1.9 | 2.0 |
| Number of click-throughs | 1.2 | 0.7 |

groups reported the same effect of the COVID-19 pandemic. More than 1 in 3 participants mentioned that the pandemic had disrupted their work. The average effect of COVID-19 on participants' workload, workhours, and stress levels was around 8 out of a maximum score of 12 (Fig 5), corresponding to somewhat increased workload and stress levels.

## Participant adherence to treatment and control interventions

Table 4 shows the degree to which study participants interacted with the Vitalk app or the mental health resource websites. Treatment group participants had a high number of interactions averaging 92 per day over the average duration of app use of 24 days. The distribution was skewed towards a lower use frequency. The average time the Vitalk app was used per day over the entire study period of 56 days was about 11 minutes and per day of use about 26 minutes. Study participants in the treatment group made frequent use of the mood meter checking it about every other day when using Vitalk. The distribution was skewed toward a lower frequency. The average mood meter score was close to a 'good' rating.

Control group participants accessed the website mental health resources infrequently–about twice over the study period of 56 days, with a distribution skewed towards 1 day. This was accompanied by about one click-through to the mental health resources offered, which was the only other measure of adherence available in this group. We did not measure time spent on resource use and mood status for the control group, because checking mood status was considered as raising self-awareness and therefore part of the intervention.

## Summary of outcomes at baseline and endline

Table 5 shows summary statistics for the 4 mental health and 2 resilience assessments at baseline and endline. Mean anxiety, depression, and loneliness scores fell into the lowest risk level during both assessments. Burnout scores were at moderate risk. Resilience was moderate to moderately high, and resilience-building activities were moderate at both times. Both control and treatment groups saw an improvement in mental health and resilience scores between the 2 assessment periods. However, the control group started at slightly lower mean mental health and higher resilience scores. While the differences between group means were statistically significant, they were within the same risk category and too small to be of clinical relevance. The DiD analysis takes the difference in baseline values between groups into account by comparing the change in each group over time. The change in risk categories and reliable change index complement the DiD estimator and provide a greater sense of the practical benefits of the Vitalk app.

**Table 5. Mental health and resilience scores from completed assessments by study period and group.**

| | Control Group | | Treatment Group | | Total | |
|---|---|---|---|---|---|---|
| Questionnaire | Baseline | Endline | Baseline | Endline | Baseline | Endline |
| **PHQ-9** | | | | | | |
| n | 510 | 634 | 429 | 496 | 939 | 1130 |
| Mean score *** | 3.67 | 2.99 | 4.29 | 2.72 | 3.95 | 2.87 |
| Standard deviation | 4.01 | 4.04 | 4.26 | 3.52 | 4.13 | 3.82 |
| Avg. no. of assessments | 2.9 | 3.3 | 3.2 | 4.6 | 3.0 | 3.9 |
| **OLBI** | | | | | | |
| n | 471 | 583 | 379 | 441 | 850 | 1024 |
| Mean score *** | 38.00 | 36.55 | 38.84 | 36.63 | 38.37 | 36.59 |
| Standard deviation | 6.18 | 6.83 | 5.74 | 6.41 | 6.00 | 6.65 |
| Avg. no. of assessments | 2.6 | 3.0 | 2.8 | 4.3 | 2.7 | 3.6 |
| **GAD-7** | | | | | | |
| n | 545 | 669 | 468 | 550 | 1013 | 1219 |
| Mean score *** | 4.24 | 2.89 | 4.73 | 2.93 | 4.46 | 2.91 |
| Standard deviation | 3.74 | 3.63 | 3.86 | 3.31 | 3.80 | 3.49 |
| Avg. no. of assessments | 3.1 | 3.6 | 3.3 | 4.9 | 3.2 | 4.2 |
| **UCLA Loneliness** | | | | | | |
| n | 433 | 531 | 84 | 106 | 517 | 637 |
| Mean score *** | 5.18 | 4.69 | 5.70 | 4.92 | 5.26 | 4.73 |
| Standard deviation | 1.52 | 1.59 | 1.45 | 1.74 | 1.52 | 1.61 |
| Avg. no. of assessments | 2.4 | 2.7 | 1.3 | 2.0 | 2.2 | 2.6 |
| **Resilience Building** | | | | | | |
| n | 422 | 521 | 128 | 117 | 550 | 638 |
| Mean score *** | 13.20 | 14.01 | 12.80 | 14.84 | 13.11 | 14.16 |
| Standard deviation | 3.11 | 3.40 | 2.62 | 2.65 | 3.00 | 3.29 |
| Avg. no. of assessments | 2.3 | 2.7 | 1.8 | 1.6 | 2.2 | 2.5 |
| **RS-14** | | | | | | |
| n | 438 | 540 | 327 | 394 | 765 | 934 |
| Mean score *** | 79.23 | 81.05 | 77.69 | 82.19 | 78.57 | 81.53 |
| Standard deviation | 10.02 | 11.36 | 10.81 | 10.57 | 10.39 | 11.04 |
| Avg. no. of assessments | 2.4 | 2.8 | 2.6 | 4.3 | 2.5 | 3.4 |

Mean score

***: A blue frame shows statistically significant differences between mean scores at baseline and endline based on within group t-tests. P-values are < 0.01 for all assessments in the control and treatment groups.

Mean score ***: Shaded cells show statistically significant differences between mean scores comparing study periods for each study group based on between group t-tests. P-values are < 0.05 for differences at baseline for all assessments and not significant at endline except for Resilience Building where baseline scores did not differ significantly between groups but endline results did.

All summary statistics are based on participants with valid baseline and endline data using all assessments completed during each assessment period. The average number of assessments taken each period varied between 1 and 5, depending on the type of assessment and was generally higher at endline.

Using published reference levels and the average score from each assessment for each study period, participants were classified according to "moderate to high" mental health risk and "very low to low" resilience levels. As seen in Table 6, fewer participants in the control and treatment groups were classified as "moderate to high" risk for mental wellbeing and "very low

**Table 6. Study participants with moderate to high risk levels for mental health or low levels of resilience.**

| Assessment and risk level | Control Group | | | | | | Treatment Group | | | | | | Sample at base- and endline | | |
|---|---|---|---|---|---|---|---|---|---|---|---|---|---|---|---|
| | Baseline | | Endline | | Total | | Baseline | | Endline | | Total | | Control Group N | Treatment Group N | Total N |
| | n | % | n | % | n | % | n | % | n | % | n | % | | | |
| Depression: moderate, moderately severe, severe | 32 | 11% | 25 | 9% | 57 | 10% | 30 | 14% | 18 | 9% | 48 | 11% | 291 | 211 | 502 |
| Burnout: moderate or high | 211 | 74% | 187 | 65% | 398 | 70% | 156 | 78% | 121 | 60% | 277 | 69% | 286 | 201 | 487 |
| Anxiety: moderate or severe | 37 | 13% | 24 | 8% | 61 | 10% | 29 | 13% | 13 | 6% | 42 | 10% | 296 | 215 | 511 |
| Loneliness: lonely | 148 | 53% | 105 | 38% | 253 | 45% | 44 | 63% | 36 | 51% | 80 | 57% | 280 | 70 | 350 |
| Resilience building: low or no activity | 49 | 18% | 44 | 16% | 93 | 17% | 14 | 16% | 2 | 2% | 16 | 9% | 278 | 85 | 363 |
| Resilience: very low, low, or on the low end | 71 | 25% | 54 | 19% | 125 | 22% | 56 | 31% | 38 | 21% | 94 | 26% | 280 | 183 | 463 |

Risk levels are based on the average assessment score per participant for each study period.

to low" resilience levels at endline than at baseline. However, decreases in depression, burnout, and anxiety, and increases in resilience- building activities and resilience were greater in the treatment than control group, consistent with the changes in mean assessment scores shown in Table 5. Loneliness did not follow this pattern, with a larger percentage of participants improving in the control than in the treatment group.

## Difference-in-differences (DiD) estimators

The DiD estimates shown in Fig 6 suggested a significant positive effect of Vitalk in reducing anxiety and depression, and in building resilience and increasing resilience-building activities. The coefficients for the interaction terms between study group and study period showed that Vitalk was more effective than the passive control intervention in reducing depression (DiD estimator -0.68 [95% CI -1.15 to -0.21]), anxiety (DiD estimator -0.44 [95% CI -0.88 to 0.01]), and burnout (DiD estimator -0.58 [95% CI -1.32 to 0.15]). Burnout was just short of statistical

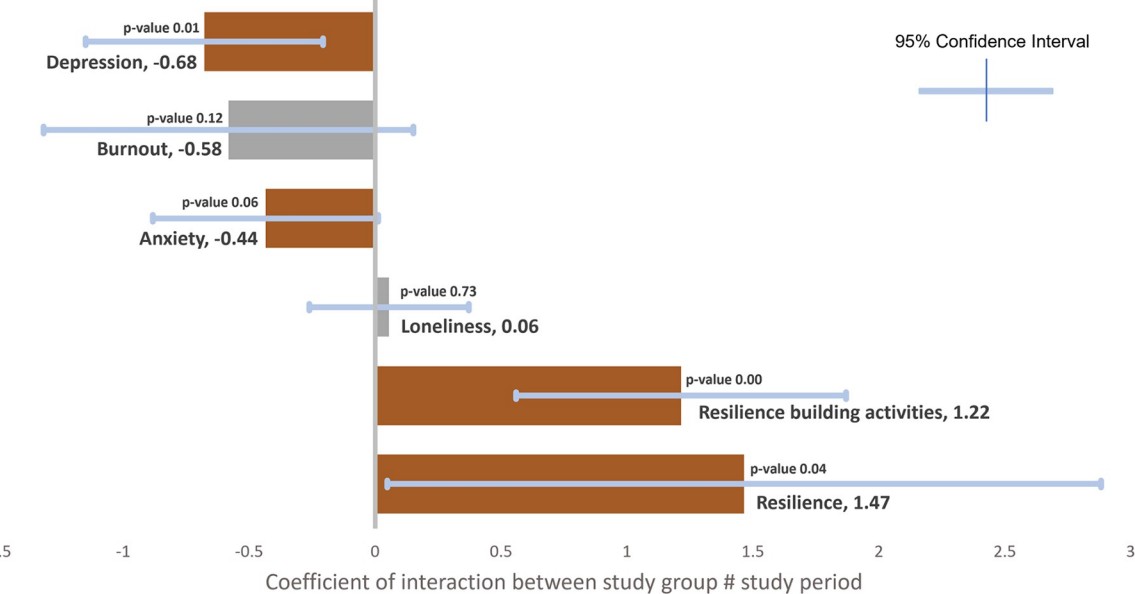

**Fig 6. Difference-in-differences estimators with 95% confidence intervals from mixed effects linear models.**

significance at the 90% level. Changes in resilience were positive and significantly greater in the treatment group (DiD estimator 1.47 [95% CI 0.05 to 2.88]). Similarly, this group showed significantly greater resilience-building activities (DiD estimator 1.22 [95% CI 0.56 to 1.87]). There seemed to exist no treatment effect on loneliness, which was omitted from the estimation of the RCI and risk level changes.

## Effect size

Although the DiD estimates showed that Vitalk use positively impacts mental health and resilience outcomes, they did not inform about the size of the effect since each outcome assessment uses a different scale. The effect size for each outcome was measured using standardized mean differences (SMD, Cohen's d). We also calculated Hedges' g–another SMD measure–to account for the unequal sample size in the treatment and control groups but do not show them separately since the values with 2 decimals were identical to Cohen's d. Fig 7 shows the results comparing the effect size for the control and treatment groups. Participants in the treatment group experienced a consistently larger effect for the mental health and resilience assessments (a medium effect), with the control group trending towards a small effect. Vitalk use had the largest impact on resilience-building activities with a large effect size (more than 3 times the effect in the control group). Effect size is calculated from statistical inference and therefore has confidence intervals (CI) shown in Fig 7. None of the 95% CI contain zero, indicating that both study groups experienced a positive effect–a decrease in depression, burnout, anxiety, and loneliness, and an increase in resilience-building activities and resilience. Possible explanations include increased self-awareness in both groups because of repeated mental health assessments, benefits of access to web resources in the control group, and a small but measurable test bias due to repeat measurements [39]. Depression and resilience-building activities showed the largest difference in effect size between the control and treatment groups, with no overlapping 95% CI.

## Reliable change index (RCI)

A comparison of the mean and median scores for mental health and resilience outcomes pre- and post-intervention in Tables 5 and 6 suggests that participants saw improved mental

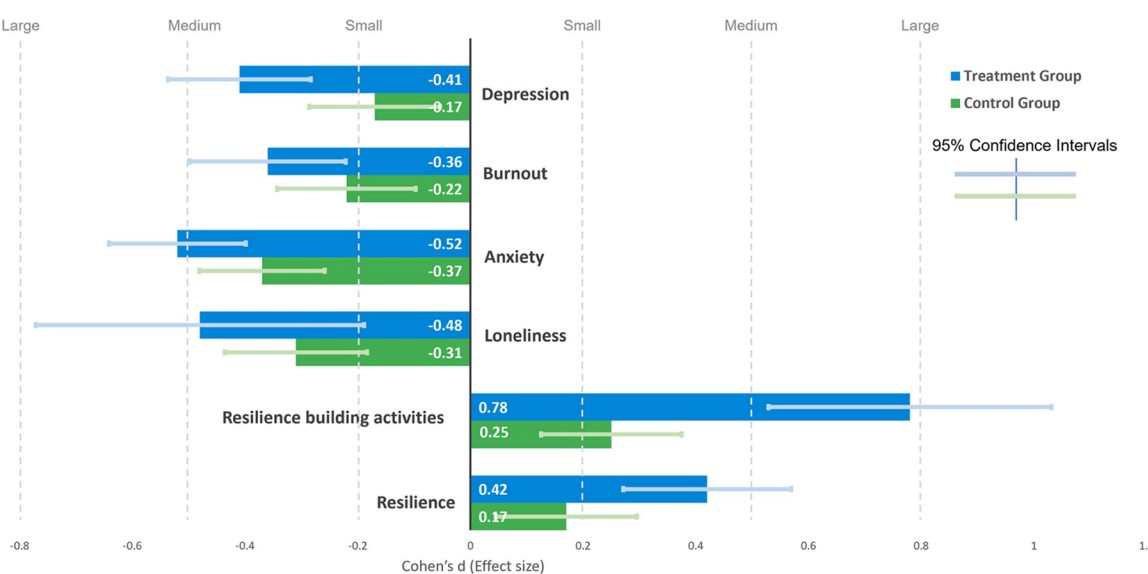

**Fig 7. Effect size post-intervention by assessment type and study group.**

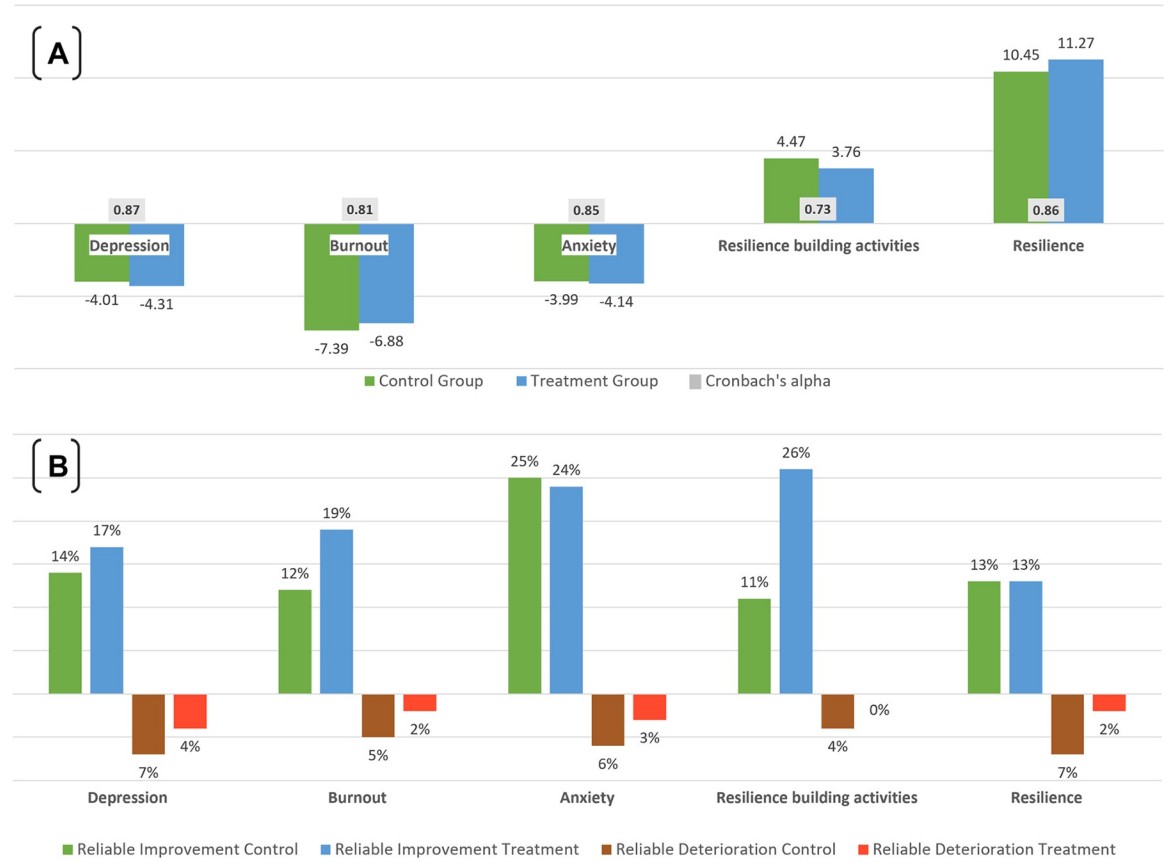

**Fig 8. Reliable change in mental wellbeing and resilience outcomes.** [A] Reliable change index (RCI). [B] Reliable improvements or deterioration in outcomes between baseline and endline.

wellbeing and resilience outcomes. However, this is the net result of a larger number of participants improving in outcomes than participants deteriorating in outcomes. To assess whether this change may have arisen by chance alone, we calculated the RCI and the proportion of participants with a change equal or greater than the RCI. The results are shown in Fig 8A and 8B. The RCI for each outcome was similar in the control and treatment groups. Reliable improvement was greater in the treatment than in the control group for depression and burnout and was about the same for anxiety. The proportion that saw a reliable increase in resilience-building activities was more than twice in the treatment than that of the control group. Reliable change in resilience was the same in both groups. While overall, fewer participants experienced a deterioration in mental health and resilience, reliable change was consistently greater in the control than in the treatment group. Fig 8A also shows Cronbach's alpha as a reliability measure for each assessment, which was used in the calculation of the RCI. Cronbach's alpha is very good (greater than 0.80) for 4 assessment instruments and acceptable for resilience-building activities.

## Mental health and resilience risk level change

While any improvement in mean scores for mental health and resilience assessments between baseline and endline is a positive outcome, a difference in means, although significant, did not inform whether these were of practical relevance. To assess whether a change in scores equated

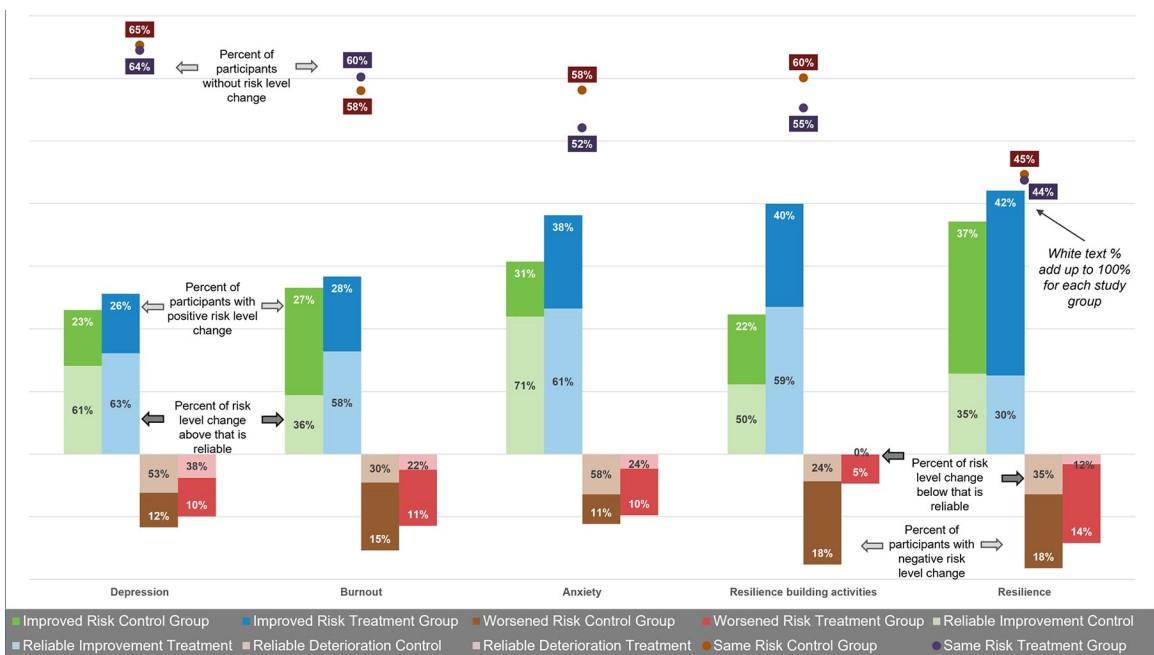

**Fig 9. Percent of participants with improved, worsened, or same mental wellbeing and resilience risk levels.** White color text percent: percent without risk level change or with one or more risk levels difference between baseline and endline. Black color text percent: percent of risk level change that is reliable based on the RCI.

to a change in risk levels, we categorized each participant's scores into risk levels using published cutoff points. We then estimated the proportion of participants changing risk categories by either moving to a lower or higher level (Fig 9). The improvement in risk levels was consistently greater in the treatment than in the control group. The effect on anxiety was greater than on depression or burnout. The depression scale showed the least amount of change, with about two-thirds of participants staying at the same risk level pre- and post-intervention. The proportion of participants increasing resilience-building activities by at least 1 level was almost twice as high in the treatment group compared to the control group. The control group consistently saw more cases with a deteriorated outcome than the treatment group. Not all the changes of mental health risk or resilience levels were statistically significant. Therefore, Fig 9 also shows the proportion of all those changing risk levels that were reliable based on the RCI. For burnout, the proportion of reliable change was considerably higher in the treatment group (58%) than in the control group (36%). For anxiety, 71% in the control group saw reliable change compared to 61% in the treatment group. Participants in the control group whose mental wellbeing and resilience outcomes deteriorated between study periods experienced a consistently greater reliable change than those in the treatment group. The proportion of participants with reliable change in Fig 9 is equivalent to the percentage of participants with reliable improvement or deterioration in Fig 8B.

## Determinants of mental health and resilience

Ordinary least squares models regressed participant characteristics on mental health and resilience outcomes using data from all participants with valid mental health and resilience assessments, controlling for study group and study period. Table 7 shows that several factors have a significant effect on several mental health and resilience outcomes. Anxiety increased by 0.03 points with each year of age. Likewise, depression increased by 0.04 points and loneliness by

**Table 7. Determinants of mental health and resilience outcomes for the control and treatment groups combined.**

| Independent variables | Dependent variables (mental health and resilience scores) | | | | | | | | | | | |
|---|---|---|---|---|---|---|---|---|---|---|---|---|
| | Depression | | Burnout | | Anxiety | | UCLA | | Resilience building activities | | Resilience | |
| | (PHQ-9) | | (OLBI) | | (GAD-7) | | LONELINESS | | | | (RS-14) | |
| | Coefficient [CI] | | Coefficient [CI] | | Coefficient [CI] | | Coefficient [CI] | | Coefficient [CI] | | Coefficient [CI] | |
| z-score of time spent @ | -0.35*** | [-0.50,-0.21] | -0.34** | [-0.62,-0.05] | -0.42*** | [-0.55,-0.28] | -0.19*** | [-0.28,-0.10] | 0.09 | [-0.09,0.27] | 0.00 | [-0.51,0.51] |
| Cadre: clinical officer † | 0.42 | [-0.28,1.11] | 1.30** | [0.06,2.55] | 0.44 | [-0.19,1.08] | -0.17 | [-0.57,0.23] | -0.40 | [-1.19,0.39] | -2.54** | [-4.74,-0.35] |
| Cadre: doctor † | -0.16 | [-2.15,1.83] | 0.64 | [-2.78,4.05] | -0.44 | [-2.31,1.43] | 1.37** | [0.26,2.48] | -2.74** | [-5.03,-0.46] | -0.93 | [-6.93,5.07] |
| Cadre: medical assistant † | -1.34*** | [-2.23,-0.45] | 1.62** | [0.03,3.21] | -1.07*** | [-1.88,-0.27] | -0.87*** | [-1.35,-0.39] | 0.57 | [-0.46,1.60] | -3.72*** | [-6.55,-0.89] |
| Cadre: pharmacist † | 0.89 | [-0.86,2.64] | 2.46 | [-0.63,5.55] | 1.16 | [-0.42,2.74] | 0.44 | [-0.41,1.28] | 0.78 | [-1.05,2.60] | -7.44*** | [-12.86,-2.01] |
| Cadre: technician † | 0.81 | [-0.85,2.47] | -1.16 | [-4.12,1.80] | 0.92 | [-0.54,2.37] | 0.35 | [-0.72,1.43] | -0.16 | [-2.37,2.04] | -1.77 | [-7.04,3.51] |
| Age (years) | 0.04*** | [0.01,0.06] | -0.04* | [-0.09,0.01] | 0.03** | [0.00,0.05] | 0.02** | [0.00,0.04] | 0.04** | [0.01,0.08] | -0.02 | [-0.11,0.06] |
| Sex ‡ | 0.18 | [-0.22,0.58] | 0.91** | [0.20,1.61] | 0.44** | [0.08,0.80] | 0.11 | [-0.11,0.33] | -0.40* | [-0.85,0.04] | -0.57 | [-1.83,0.69] |
| Care level: secondary $ | 0.75** | [0.10,1.40] | 1.47** | [0.31,2.64] | 0.74** | [0.15,1.33] | -0.05 | [-0.40,0.30] | 0.37 | [-0.32,1.05] | -0.30 | [-2.37,1.77] |
| Care level: tertiary $ | 1.05*** | [0.49,1.61] | 2.02*** | [1.01,3.04] | 1.01*** | [0.50,1.52] | 0.06 | [-0.27,0.39] | 0.47 | [-0.20,1.15] | -1.21 | [-3.04,0.61] |
| Facility type: private nfp !! | -0.62* | [-1.27,0.03] | -0.10 | [-1.27,1.06] | -0.60** | [-1.19,0.00] | 0.25 | [-0.08,0.58] | -0.25 | [-0.92,0.42] | -2.09** | [-4.13,-0.04] |
| Facility type: private fp !! | 0.79* | [-0.07,1.65] | 2.07*** | [0.53,3.61] | 0.49 | [-0.28,1.26] | 0.96*** | [0.48,1.45] | -0.62 | [-1.58,0.33] | -2.47* | [-5.17,0.22] |
| Location: peri-urban Ŧ | 0.49 | [-0.17,1.16] | -0.18 | [-1.36,1.01] | 0.48 | [-0.12,1.07] | 0.02 | [-0.39,0.43] | 0.04 | [-0.75,0.84] | -0.68 | [-2.81,1.44] |
| Location: rural Ŧ | 0.53* | [-0.01,1.06] | -0.20 | [-1.16,0.76] | 0.31 | [-0.18,0.81] | -0.08 | [-0.37,0.22] | 0.04 | [-0.57,0.65] | 1.50* | [-0.21,3.22] |
| Service: counseling § | -1.62*** | [-2.65,-0.58] | 1.15 | [-0.75,3.04] | -1.02** | [-1.96,-0.08] | 0.17 | [-0.48,0.81] | -1.19* | [-2.45,0.07] | 0.33 | [-3.05,3.72] |
| Service: inpatient care § | -1.20*** | [-2.00,-0.40] | -0.05 | [-1.51,1.41] | -1.04*** | [-1.76,-0.32] | -0.47** | [-0.91,-0.02] | 0.24 | [-0.68,1.16] | -2.44* | [-5.02,0.14] |
| Service: intensive care § | -1.42*** | [-2.30,-0.54] | 0.65 | [-0.94,2.25] | -0.89** | [-1.68,-0.10] | -0.05 | [-0.56,0.45] | -0.21 | [-1.24,0.82] | -0.91 | [-3.75,1.93] |
| Service: maternity care § | -0.92** | [-1.64,-0.21] | 1.45** | [0.15,2.75] | -0.55* | [-1.20,0.10] | -0.41** | [-0.81,-0.02] | 0.10 | [-0.72,0.91] | -1.72 | [-4.02,0.59] |
| Service: support & diagnostics § | -1.69** | [-3.34,-0.03] | -0.72 | [-3.69,2.25] | -1.36* | [-2.81,0.10] | -0.89* | [-1.93,0.14] | -0.28 | [-2.43,1.87] | 3.69 | [-1.65,9.03] |
| Service: specialty care § | -2.06*** | [-2.94,-1.19] | 0.73 | [-0.86,2.32] | -1.60*** | [-2.40,-0.81] | -0.76*** | [-1.25,-0.27] | -0.53 | [-1.55,0.49] | 0.24 | [-2.57,3.05] |
| Service: surgery care § | -1.53*** | [-2.48,-0.58] | 0.88 | [-0.83,2.58] | -1.47*** | [-2.33,-0.60] | -1.00*** | [-1.52,-0.47] | -0.91* | [-1.97,0.16] | -1.00 | [-4.04,2.04] |
| Smartphone experience score | 0.06*** | [0.03,0.09] | 0.05* | [-0.01,0.11] | 0.08*** | [0.05,0.11] | 0.03*** | [0.01,0.05] | 0.04** | [0.00,0.08] | -0.09 | [-0.19,0.02] |
| History of MH counseling | 0.64*** | [0.42,0.86] | 0.35* | [-0.05,0.76] | 0.36*** | [0.17,0.56] | -0.03 | [-0.18,0.11] | -0.07 | [-0.39,0.24] | 0.27 | [-0.46,0.99] |
| COVID-19 impact score | 0.02 | [-0.04,0.08] | 0.20*** | [0.09,0.31] | 0.05* | [0.00,0.11] | 0.01 | [-0.02,0.05] | -0.13*** | [-0.20,-0.06] | 0.22** | [0.02,0.41] |
| COVID-19 ability to work impact | 0.62*** | [0.43,0.81] | 0.95*** | [0.61,1.29] | 0.38*** | [0.21,0.55] | 0.16*** | [0.05,0.27] | -0.24** | [-0.47,-0.01] | -0.10 | [-0.71,0.50] |
| History of chat app use for MH | -0.02 | [-0.17,0.13] | -0.24* | [-0.51,0.04] | -0.07 | [-0.21,0.07] | -0.06 | [-0.15,0.03] | 0.28*** | [0.10,0.46] | 0.19 | [-0.32,0.69] |
| History of website use for MH | 0.20*** | [0.05,0.35] | -0.07 | [-0.34,0.20] | 0.17** | [0.04,0.31] | 0.05 | [-0.03,0.14] | 0.06 | [-0.11,0.24] | 0.30 | [-0.19,0.78] |

*(Continued)*

**Table 7.** (Continued)

| Independent variables | Dependent variables (mental health and resilience scores) | | | | | | | | | | | |
| --- | --- | --- | --- | --- | --- | --- | --- | --- | --- | --- | --- | --- |
| | Depression | | Burnout | | Anxiety | | UCLA | | Resilience building activities | | Resilience | |
| | (PHQ-9) | | (OLBI) | | (GAD-7) | | LONELINESS | | | | (RS-14) | |
| | Coefficient [CI] | | Coefficient [CI] | | Coefficient [CI] | | Coefficient [CI] | | Coefficient [CI] | | Coefficient [CI] | |
| Treatment group ⸗ | 0.56*** | [0.20,0.93] | 0.43 | [-0.23,1.09] | 0.63*** | [0.30,0.97] | 0.61*** | [0.34,0.87] | 0.04 | [-0.45,0.54] | -0.46 | [-1.64,0.72] |
| Study period mid, end, post ≡ | -0.60*** | [-0.76,-0.43] | -0.95*** | [-1.25,-0.65] | -0.80*** | [-0.95,-0.65] | -0.27*** | [-0.36,-0.17] | 0.59*** | [0.40,0.78] | 1.63*** | [1.09,2.17] |

[CI]: [95% Confidence Interval]; P-values

* ≤ 0.10

** ≤ 0.05

*** ≤ 0.01, probability that coefficient values are 0 based on multivariable ordinary least squares regression

z-score of time spent @ stands for standardized hours spent on Vitalk for the treatment group, and standardized number of days with web resources accessed for the control group

Reference categories: † Cadre: nurse, ‡ Male, $ Care level: primary, !! Facility type: public, T̄ Location: urban, § Service: outpatient care, ⸗ Control group, ≡ Baseline

Abbreviations: nfp = not-for-profit, fp = for profit, MH = mental health, mid = Midline, end = Endline, post = Post-endline period

0.02 points. Burnout appeared to decrease with age by -0.04 points. Resilience-building activities increased with age by 0.04 points. Participant gender was negatively associated with anxiety and burnout, with women having worse outcomes than men by 0.44 points and 0.91 points, respectively. In addition, women showed less resilience-building activities of -0.40 points than men. While this was consistent with women having -0.57-point lower resilience levels, this was not statistically significant at the 0.05 level for both variables. Age and gender were also closely correlated with smartphone use experience (not shown in Table 7). With each year increment of age, smartphone experience decreased by -0.17 points [95% CI -0.17 to -0.15]; and females saw -0.99 point [95% CI -1.15 to -0.82] lower smartphone experience than men.

Secondary care providers had significantly worse outcomes (higher scores) for anxiety, depression, and burnout compared to primary care providers, ranging from 0.74 points for anxiety to 1.47 points for burnout. Tertiary care providers scored 2.02 points higher for burnout than primary care providers. Outpatient care saw consistently higher anxiety, depression, and loneliness than any of the other types of care, with surgery (depression -1.53 points) and specialty care (depression -2.06 points) having the largest difference. Maternity care was the only type associated with a significantly 1.45-point higher burnout than outpatient care. Medical assistants scored significantly lower for anxiety, depression, and loneliness compared to nurses, but they had 1.62-point higher burnout, and -3.72-point lower resilience than the latter.

Anxiety, depression, burnout, and loneliness outcomes all decreased with the z-scores of time spent with the treatment or control interventions, ranging from -0.19 points for loneliness, to -0.42 points for anxiety. Conversely, greater use of smartphone features was associated with significantly higher levels of anxiety, depression, burnout, and loneliness, ranging from 0.04 points for loneliness to 0.08 points for anxiety. Participants with greater smartphone experience had higher resilience-building activities by 0.04 points. While smartphone use showed a -0.09-point lower level of resilience, this was not statistically significant (p-value of 0.104).

Participants with a history of mental health counseling had higher levels of anxiety (0.36 points), depression (0.64 points), and burnout (0.35 points). The COVID-19 pandemic and its impact on work-related stress, workload, and working hours increased participants' anxiety by

0.05 points and burnout by 0.20 points. The coefficients for depression and loneliness also suggest a negative effect, but they were smaller and not statistically significant. While resilience levels increased by 0.22 points with the impact of COVID-19, resilience-building activities were affected negatively by -0.13 points. We observed a similar but stronger effect on participants' ability to work during the pandemic for all mental health outcomes and resilience-building activities, but not for resilience levels, ranging from 0.16 points for loneliness, to 0.95 points for burnout. Age and gender were significantly associated with both COVID-19 related variables, but the effect was small (odds ratios close to 1). A history of using a chat app for mental health significantly increased resilience-building activities by 0.28 points. Participants who had previously accessed Internet resources for mental health saw significantly greater anxiety (0.17 points) and depression (0.20 points).

While there were differences in outcomes between control and treatment groups, only higher anxiety, depression, and loneliness were significantly associated with the treatment group. As expected from the DiD, effect size, and reliable change analyses, there was a significant relationship between study period and all mental wellbeing and resilience outcomes with strong gains post-intervention.

A similar ordinary least squares model for the treatment group regressed participant characteristics on mental health and resilience outcomes while controlling for study group and study period. Table 8 shows independent variables unique to the treatment group only. Mood scores were positively correlated with all mental health outcomes and resilience levels but not with resilience-building activities. A 1-point increase in mood status was associated with a -0.83 point decrease in anxiety, -0.71 point decrease in depression, -1.44 point decrease in burnout, -0.38 point decrease in loneliness, and 2.37 point increase in resilience. Specific carelines selected by participants had little impact on mental health or resilience outcomes, except when comparing the stress and COVID-19 carelines to the anxiety careline. The stress careline was associated with significantly greater depression but lower burnout than the anxiety careline; there was no significant relationship between the stress careline and other outcomes. The COVID-19 careline was associated with lower mental wellbeing scores for all 4 assessments and higher resilience-building activities and resilience levels. However, these relationships only trended towards statistical significance at the 0.1 level for depression and loneliness.

**Table 8. Additional determinants of mental health and resilience outcomes for the treatment group.**

| | Dependent variables (mental health and resilience scores) | | | | | | | | | | |
|---|---|---|---|---|---|---|---|---|---|---|---|
| | Depression | | Burnout | | Anxiety | | UCLA | | Resilience building activities | | Resilience |
| | (PHQ-9) | | (OLBI) | | (GAD-7) | | LONELINESS | | | | (RS-14) |
| Independent variables | Coefficient [CI] | | Coefficient [CI] | | Coefficient [CI] | | Coefficient [CI] | | Coefficient [CI] | | Coefficient [CI] | |
| Mood score | -0.71*** | [-0.95,-0.47] | -1.44*** | [-1.85,-1.03] | -0.83*** | [-1.04,-0.61] | -0.38*** | [-0.60,-0.17] | 0.26 | [-0.12,0.64] | 2.37*** | [1.59,3.16] |
| Careline: depression † | 0.20 | [-0.83,1.22] | -1.34 | [-3.10,0.41] | -0.26 | [-1.17,0.65] | -0.09 | [-1.13,0.94] | -1.18* | [-2.56,0.20] | -3.17* | [-6.45,0.11] |
| Careline: stress † | 1.06** | [0.21,1.92] | -1.79** | [-3.25,-0.34] | 0.62 | [-0.14,1.38] | -0.36 | [-1.31,0.59] | -0.02 | [-1.26,1.22] | -0.86 | [-3.69,1.97] |
| Careline: relationships † | -0.32 | [-1.16,0.51] | -1.03 | [-2.50,0.44] | 0.36 | [-0.39,1.10] | -0.15 | [-0.96,0.66] | -0.14 | [-1.28,0.99] | -1.88 | [-4.71,0.96] |
| Careline: COVID-19 † | -0.60* | [-1.28,0.09] | -0.82 | [-2.00,0.36] | -0.21 | [-0.83,0.40] | -0.60* | [-1.30,0.09] | 0.40 | [-0.70,1.50] | 0.36 | [-1.89,2.62] |

P-values

* ≤ 0.10

** ≤ 0.05

*** ≤ 0.01

Reference category: † Careline: anxiety

## Participant experience

At the end of the study, 820 participants completed an anonymous online participant experience questionnaire (37% from the control group and 63% from the treatment group). The gender distribution was the same (37% male and 63% female). Respondents were predominantly young (54% in the 18-to-29-year age group). Nurses made up the majority (68%). These respondent characteristics are similar to the attributes of the study population.

Fifty percent of participants from the treatment group reported using the app for more than 28 days out of the total study period of 56 days (in line with an average of 24 days of use based on app data). When asked why the app was not used more frequently, the top 3 responses were: "I did not have the time to use the app" (45%); "The app took too long to work with" (30%); and "The app was too difficult to use" (13%). Overall, 91% of users found the app easy to use; 87% found the app content relevant; 92% derived benefits from the app; and 83% felt more resilient to everyday challenges. However, Vitalk users encountered several difficulties: 32% found the trial welcome email confusing; 27% found it difficult to download and install the app; 20% found it difficult to sign into the app; 19% had difficulties entering the trial ID; 25% found getting to the mental health assessments confusing; and 33% noted that the app took too long to work with. The top 3 likes of app use were: interactions with Viki (71%); the exercises (58%); and taking the mental wellbeing and resilience assessments (55%). The top three dislikes or challenges of app use were: exposure to new content every or every other day (36%); the mental health exercises (28%); and completing the mental health and resilience assessments (25%). The net promotor score (NPS) [40], was assessed by asking how likely users were on a scale from 1 (not likely) to 10 (very likely) to recommend Vitalk to others. Sixty-six percent out of 515 respondents rated Vitalk a 9 or 10 (promoters), 23% a 7 or 8 (passives), and 11% a 6 or less (detractors). This resulted in a NPS–promoters minus detractors–of 55.

Fifty-two percent of participants from the control group reported using the Internet resources for more than 15 days out of the total study period of 56 days, a finding that is much higher than the one gleaned through web analytics (an average of less than 2 days accessing web resources). When asked why the Internet resources were not used more frequently the top 3 responses were: "I did not have the time to use the resources" (46%); "The resources took too long to work with" (34%); and "The content was too difficult to understand" (13%). Overall, 87% of users found the Internet resources easy to use; 92% found the resource content relevant; 95% derived benefits from the resources; and 87% felt more resilient to everyday challenges. However, Internet resource users encountered several difficulties: 17% found the trial welcome email confusing; 22% found it difficult to access the resources; 11% found it difficult to get to the website; 7% had difficulties entering the trial ID; 11% found getting to the mental health assessments confusing; and 43% noted that the resources took too long to work with. The top three likes of web resource use were: taking the mental wellbeing and resilience assessments (61%); reading new content (55%); and the exercises (52%). The top three dislikes or challenges of resource use were: reading the website content (44%); finding new content by following links (36%); and doing mental health exercise (28%). Twenty-three percent found completing the mental health and resilience assessments challenging. When asked how likely users were on a scale from 1 (not likely) to 10 (very likely) to recommend the Internet resources to others, 64% out of 305 respondents rated Internet resources a 9 or 10 (promoters), 23% a 7 or 8 (passives), and 13% a 6 or less (detractors). The resulting NPS of 51 is similar to the treatment group.

## Discussion

Based on our literature review, this study is the first RCT of a mental health chat app for health workers combining multiple mental health and resilience outcomes. It was the first to include a measurement of resilience and to explore the potential of a mental health chat app to stimulate resilience-building activities among its users. The RCT took place in a low-resource setting in Southern Africa with very limited access to mental health counseling and therapy. The challenges that Malawi faces meeting population health needs have been further worsened by the COVID-19 pandemic.

The strengths of our RCT were the recruitment of more than 1,500 health professionals, and the management team's continuous encouragement of participants over the 8-week study period. Participants were successfully randomized into the 2 study arms while maintaining single blinding of the research team. The overall enrollment rate was 94%, with group-specific retention rates of 29% (treatment group) and 40% (control group) (see CONSORT diagram in Fig 1). The numerator for retention rates is the number of participants who completed base- and endline assessments for GAD-7, the denominator is the number of participants allocated to either the control or treatment group. Compared to other much smaller RCTs of i-CBT, retention rates were high due to weekly reminders through WhatsApp and email messages, as well as instant support of users meeting issues during registration, sign in or app use. A trial of adolescents with self-identified anxiety concern (with fewer than 50 participants per group) reported 8-week retention rates of 28% (treatment group) and 58% (control group) [41]. An RCT of i-CBT of participants with depression and anxiety symptoms referred to care (substantially different from our study population) reported much higher 8-week retention rates of 82% (treatment group) and 72% (control group) [42].

The results from the DiD estimators obtained from mixed linear effect modeling support our working hypothesis that use of a chatbot such as Vitalk can effectively improve mental health and resilience outcomes among health professionals in Malawi. The coefficient for the interaction terms between study group and assessment period varied between -0.68 for depression, -0.58 for burnout, -0.44 for anxiety, 1.22 for resilience building activities, and 1.47 for resilience levels, indicating that active engagement with the app was more effective in improving mental wellbeing (except loneliness) and increasing health worker resilience compared to the passive offering of Internet resources. We are assured that Vitalk app use causes greater improvements in mental wellbeing and resilience than the alternative due to the experimental study design and the size of the DiD estimators and confidence intervals (most do not include zero or absence of a treatment effect). It should be noted that such a strong treatment effect was observed despite considerable technical issues faced by Vitalk users. These challenges likely moderated the effect size as indicated by the disproportionate attrition in the treatment group and the feedback from the anonymous user experience survey, suggesting that this was a very realistic trial of a constantly evolving innovative mobile technology.

Based on DiD estimators, Vitalk use did not appear to be associated with changes in loneliness. Possible reasons include: a small effective sample for the treatment group (n = 70); a coarse 3-question instrument that may not be sufficiently sensitive for measuring the effect of Vitalk's relationship careline; or app use in general. Although we explored behavioral theory about loneliness, we could not find any studies that examined whether i-CBT or chatbot use influenced social isolation. However, a recent meta-analysis by Bryan et al. found that workplace loneliness was significantly associated with burnout and other health and personal factors [43].

Our findings add substantial evidence to existing research. Unlike other published studies, we tested the effectiveness of Vitalk by implementing a large RCT in a LMIC context where access to mental wellbeing and resilience support is extremely limited. Compared to our

sample sizes of 200 participants or more per study arm for most of our outcome measures, the other existing RCTs tend to suffer from very small samples of about 30 individuals or fewer. Our RCT is also the first of its kind to be applied to the health workforce, which during normal times is at very high risk for work-related stress and burnout, but even more so during the COVID-19 pandemic. In addition to mental wellbeing outcomes, our study not only measured levels of resilience but also resilience-building activities by participants. We were not able to find studies that looked at resilience as an outcome to compare them to our findings.

While both study groups experienced improvements in mental wellbeing and resilience between baseline and endline, the effect size was consistently larger for the treatment than for the control group, further supporting our working hypothesis that Vitalk is more effective in improving mental health and resilience outcomes than passive Internet resources. While we were not able to compare our DiD results to the published literature, we can compare the effect size, which is commonly reported for RCTs. Our findings suggest an approximately medium effect size, varying between $d = -0.36$ to $-0.52$ for mental health, and $d = 0.42$ to $0.78$ for resilience, which is similar to reported literature on the effectiveness of c-CBT of depression and anxiety based on 24 RCTs with sample sizes between 19 and 257 adolescents and young adults with symptoms of anxiety or depression [11]. However, it is smaller than an average effect size of $g = 0.77$ (95% CI 0.59–0.95) reported from a meta-analysis of 49 RCTs [9]. An RCT (n = 56) on the use of Woebot (a chatbot) to deliver CBT to young adults with depression and anxiety reported a significant reduction of both outcomes with an effect size of $d = 0.44$ and $0.37$, respectively [44]. An RCT of the use of the Vivibot chatbot (n = 45) by young people with cancer found a significant interaction effect between time and study group of -0.41 for reducing anxiety, but no effect on depression [45]. A review of 12 papers cited weak evidence of the ability of chatbots to reduce depression, with an effect size of -0.55 (95% CI -0.87 to -0.23), which is very similar to our result, but reported outcomes for anxiety contradicting our findings [19]. Effect size may be influenced by the characteristics of the study population. While our study targeted healthy healthcare workers, other studies enrolled participants with predetermined mental health issues or underlying illnesses. The fact that healthcare workers have self-selected to cope with stressful situations compared to participants from the general population, or those clinically diagnosed with symptoms, may explain some of the discrepancies in effect size.

Nonexperimental studies showed a greater effect size than RCTs. An earlier effectiveness study of self-selected Vitalk users from the general population reported a large effect size of $d = 0.81$ and greater for mental health outcomes [16]. Another uncontrolled trial of clients with severe symptoms of depression and anxiety reported much higher scores for depression and anxiety (based on PHQ-9 and GAD-7 assessments) than our study, as well as effect sizes ranging from $d = 0.61$ to $0.83$ [46].

The LMIC setting of our RCT may account for a smaller effect size than anticipated compared to results from studies in middle- and high-income countries. Limited and expensive access to mobile phones in Malawi offers less opportunity for self-help and other innovative app use among the population, which may have constrained participants' ability to benefit from the full potential of a mental health chatbot. This is supported by the fact that a high proportion of participants reported that they did not have the time to use the app or Internet resources (1 in 5) or found that it took too long to work with them (1 in 3). This could also imply that participants tried to conserve precious airtime. Large differences in mean scores between our study and others suggest that culture may also play an important role in shaping how participants respond to mental health and resilience questionnaires [47, 48]. For example, our RCT in Malawi reported mean anxiety scores (4.5 at baseline and 2.9 at endline) and depression scores (4.0/2.9) that were less than half of those found in Brazil (pre-/post-app use means of 12.2/8.1 for anxiety, and 15.9/10.4 for depression) [16].

Many of the improvements in mental health and resiliency scores documented in our study are of statistical and clinical relevance and are welcome outcomes. Changes in risk levels were based on RCIs of 4 for the GAD-7 (anxiety) and PHQ-9 (depression) calculated from our sample. These are similar to or lower than those reported by other studies (e.g., an RCI of 6 for PHQ-9 and an RCI of 4 for GAD-7 [46], and an RCI of 6 for GAD-7 [49]). Based on our RCIs, we estimate that about one- to two-thirds of the improvements in both study groups were reliable (see Fig 9). Applying the RCI, we saw reliable improvements for depression of 14% (control group) and 17% (treatment group); and improvements in anxiety of about 25% (both groups). This is considerably lower than the reliable change in clinical caseness reported by other studies (e.g., 52% for anxiety and 45% for depression [16] and between 58% and 70% across assessments [46]). These discrepancies are likely due to differences in study populations (randomly selected health workers versus self-selected general population), with much lower mean scores for our participants, and a small proportion classified at moderate or severe risk levels of about 13% at baseline and 8% at endline for depression and anxiety, respectively. A small proportion (0% to 7%) of participants saw a reliable worsening of mental health and resilience scores; the effect was consistently higher in the control group. This decline is in line with about 2% to 5% reported by Daley et al. [16].

While controlling for trial group affiliation and study period, our analysis explored which user or engagement characteristics most significantly explained the outcomes of interest. Although these factors may vary by country context, they must be considered when planning mental health support strategies and helping health workers increase their resilience to meet everyday challenges. Engagement with the app or the website, measured as z-scores of time spent, was strongly correlated with reduced anxiety, depression, burnout, and loneliness. Gan et al. [50] also found a significant association between engagement in digital mental health interventions and mental health improvements. A systematic review by Molly and Anderson [51] identified several studies that showed statistically significant associations between engagement and clinical improvement. Our data did not indicate that time was a determinant of resilience or resilience-building activities.

Our study identified a need for user interface and app processes to be designed for people with less experience using smartphones and online resources and with a higher risk of adverse mental health outcomes. In our study, we found that older and female participants tended to have greater anxiety, depression, loneliness (age only), and burnout (gender only), as well as less experience with smartphone and Internet use. In a study of the health workforce in Jordan, age and gender were associated with worse mental health outcomes for males [52]. A rapid scoping review of occupational stress, burnout, and depression in women in healthcare during the COVID-19 pandemic showed that female health workers were at increased risk [53].

Compared to those working in other types of medical and ancillary services, levels of anxiety and depression appear to be highest among our participants working in outpatient care, a service in which healthcare workers may be required to pay special attention to mental health support, but, according to anonymous user feedback, they may also face the greatest time constraints for accessing online mental health solutions. Participants working in secondary and tertiary care institutions were at higher risk for anxiety, depression, and burnout compared to those in primary care, a finding similar to a study in Peru [54]. In our study, we found that public facilities were associated with greater mental health risks than private, not-for-profit facilities. Public facilities may be able to draw upon the strategies used by private facilities to reduce anxiety and depression. However, the public sector in Malawi seems better equipped to build health worker resilience compared to the private sector. Work location (urban versus rural), was not significantly associated with the outcomes (at the 95% level).

Our findings that current or past exposure to mental health counseling was associated with greater anxiety, depression and burnout confirmed those of Varma et al. [55]. Our participants saw a negative impact of the COVID-19 pandemic across all outcomes (except for resilience scores), similarly to findings from a study in Jordan [52]. A study of healthcare and allied workers involved in the COVID-19 pandemic from India found high levels of severe and extremely severe anxiety (23.2%) and severe and extremely severe depression (11.4%) [56]. The level of severe anxiety was almost double that found in our study (which did not focus on workers involved in the pandemic). The levels of severe depression were similar to our findings. Although the Vitalk app demonstrated no effect on loneliness, the pandemic itself had a negative impact on loneliness overall. This could be expected based on the results of a meta-analysis reporting a small but significant effect size of $g = 0.27$ of the COVID-19 pandemic on loneliness [57].

## Limitations

Our study had several limitations. Due to technical issues with the app, the response rate for the treatment group was substantially lower than for the control group. The attrition rate was greater for the loneliness and resilience-building activity assessments, reducing the power of our statistical analysis. Engagement with the app was an average of 11 minutes per day, including the completion of assessments, and was within a range of 4 to 22 minutes (median) reported by a study of user engagement with mental health apps [58]. The RCT provided participants with airtime to use Vitalk or to peruse the web resources on their mobile phones, which no doubt incentivized app use. However, this did not constitute a bias, because it affected both study groups equally. The financial support allowed us to test a mental wellbeing digital intervention in a resource-poor environment in anticipation of a future rapid increase of affordable cellular data access in sub-Saharan Africa. Our study was not representative of the entire health workforce in Malawi, because the interventions required English language proficiency and smartphone ownership, limiting the study to those with a diploma or at least 2 years of college education. Moreover, participants were predominantly from 2 urban districts within major population centers and only 18% from rural health facilities. Given that the classification of mental health and resilience relied entirely on self-assessments, which can be influenced by sociocultural factors unique to a country or study population, this phenomenon must be considered when comparing our findings to those of other studies.

As indicated by NPS scores, user satisfaction with Vitalk and web resources was high in the treatment (NPS of 55) and control (NPS of 51) groups. While we did not find NPS benchmarks for chat apps or i-CBT, a study by Marler [59] compared the NPS in an RCT of 2 smoking cessation apps and found values ranging between 1 (control app) and 57 (intervention app). A study of community mental health services found an NPS of 19, which varied considerably by respondent characteristics (from 10 for older, to 37 for younger patients) [29]. Nonetheless, given the limitations of validity and reliability of single item measurements such as the NPS [60], the magnitude of the NPS and difference between groups should be viewed with caution. Moreover, apps such as Vitalk are constantly updated and enhanced as the technology matures. During the RCT in Malawi, a newer Vitalk version with advanced artificial intelligence and natural language processing features offering a potentially better user experience was already under development in Brazil.

## Conclusions

Given Vitalk's demonstrably positive effect on mental wellbeing and resilience, it has exciting potential to be an effective tool in the arsenal of interventions to reduce work-related stress

and burnout. Based on our completion rate (1 in 3 study participants), there appears to be substantial demand for such an app and willingness to engage with it for at least 10 minutes daily, a length of time that seems adequate for most users to improve mental wellbeing and resilience, in addition to extra time for occasional assessments. User feedback suggests that several factors encourage chatbot use including natural interaction with the app; frequently updated content and exercises; and carelines adapted to 2 diverse types of users–those needing more general psychosocial support and those facing more pronounced mental health issues. Users indicated that they enjoyed the interaction with the virtual assistant, Viki, and appreciated the assessments on mental wellbeing and resilience as important benchmarks.

To mitigate the technical difficulties encountered meticulous planning of the app rollout is critical. This includes ensuring local access to an app that was developed for a different region (Brazil) and not underestimating the amount of handholding necessary to install the app on smartphones for a study population that is not a frequent user of mobile apps because of costs. Our study management team spent considerable effort supporting users online, through WhatsApp groups, and individual email support. Making the assessments less tedious and more entertaining is an ongoing challenge for app developers. The addition of elements to a life app, user identification and 2 assessments in our case, requires sufficient lead time to ensure smooth operation. Factors to consider but are beyond the influence of the study included intermittent internet access, which affected both study groups equally and limited smartphone app use experience which affected the treatment group more than the control group. App developers should weigh the advantages and disadvantages of changing the clinical risk classifications based on standardized mental health and resilience assessments from their original to fixed groupings, even if this makes sense from a user-friendliness perspective.

Vitalk was developed in Brazil and was translated from Portuguese into English and adapted to the African context to resonate with Malawian users. While it would be cost effective to make future version available more widely globally, African countries could take ownership and create an app that is well adapted to the African context by developing their own Vitalk version through a collaborative effort to make the most efficient use of their resources. These resources include not only app developers, which are widely available in Africa, but also psychologists and mental health therapists to develop and validate the app content.

While apps such as Vitalk can help health professionals cope with daily stressors and improve mental wellbeing, employers need to create less stressful work environments to maximize their effectiveness. Proven interventions to strengthen health systems function and to increase healthcare provider satisfaction include ensuring: essential supplies and equipment; an infrastructure that meets the needs of patients and staff; and adequate and well-trained human resources for health. As Internet access through mobile devices becomes more widespread and affordable in LMICs, employers in the public and private sector should consider providing free app subscriptions to their employees, organizing group sessions for mental wellbeing exercises, and encouraging app use.

Similarly to other studies, our RCT confirmed that mobile chat apps are one effective tool for improving mental wellbeing and resilience among its target audience. For the first time, our findings show that apps such as Vitalk can support the mental wellbeing and resilience of health workers to meet the challenges unique to their profession, especially in a resource poor environment where access to mental health services is very limited. Countries employ thousands of health workers, and many need support: as many as 1 in 8 health workers in our RCT needed help with anxiety or depression; 3 in 4 faced some degree of burnout; and 1 in 4 had low resilience. As countries struggle to increase access to mental health professionals, apps such as Vitalk have potential to fill the gaps in support. While the study took place during the COVID-19 pandemic, which was not foreseen when we planned the study, it provides a long-

term solution for a workforce exposed to a high-pressure environment during normal times and especially during regularly occurring emergency situations. In the aftermath of the ongoing COVID-19 pandemic, healthcare providers in the public and private sectors should consider offering access to mental wellbeing apps to all their employees to cope with everyday challenges and to prepare for future public health emergencies.

## Disclaimer

The views and opinions expressed in this paper are those of the authors and not necessarily the views and opinions of the United States Agency for International Development.

## Supporting information

**S1 File. CONSORT, 2010, checklist of information to include when reporting a randomized trial.**
(PDF)

**S2 File. Study protocol–for publication.**
(PDF)

**S3 File. Questionnaires.**
(PDF)

**S4 File. Codebook and deidentified dataset.**
(XLSX)

**S5 File.**
(DOCX)

## Acknowledgments

The authors would like to thank the management and health professionals from health centers and hospitals in Malawi for their support and participation in this study. We gratefully acknowledge Elizabeth Leitman (Chemonics International) for editing the manuscript.

## Author Contributions

**Conceptualization:** Eckhard Kleinau, Tilinao Lamba, Wanda Jaskiewicz, Ines Hungerbuehler, Demoubly Kokota, Limbika Maliwichi, Edister Jamu, Alex Zumazuma.

**Data curation:** Katy Gorentz, Donya Rahimi, Raphael Mota.

**Formal analysis:** Eckhard Kleinau.

**Investigation:** Demoubly Kokota, Limbika Maliwichi, Edister Jamu, Alex Zumazuma.

**Methodology:** Eckhard Kleinau, Tilinao Lamba, Katy Gorentz, Ines Hungerbuehler, Demoubly Kokota, Limbika Maliwichi, Edister Jamu, Alex Zumazuma.

**Project administration:** Tilinao Lamba, Katy Gorentz, Donya Rahimi.

**Software:** Mariana Negrão, Raphael Mota, Yasmine Khouri, Michael Kapps.

**Supervision:** Tilinao Lamba, Katy Gorentz, Mariana Negrão, Michael Kapps.

**Visualization:** Eckhard Kleinau.

**Writing – original draft:** Eckhard Kleinau.

**Writing – review & editing:** Eckhard Kleinau, Tilinao Lamba, Wanda Jaskiewicz, Katy Gorentz, Demoubly Kokota, Limbika Maliwichi, Edister Jamu, Alex Zumazuma.

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
