## [Decision Letter · Decision Letter 0]

6 Dec 2023

PONE-D-23-02103Effectiveness of a chatbot in improving the mental wellbeing of health workers in Malawi during the COVID-19 pandemic: A randomized, controlled trialPLOS ONE

Dear Dr. Kleinau,

Thank you for submitting your manuscript to PLOS ONE. After careful consideration, we feel that it has merit but does not fully meet PLOS ONE’s publication criteria as it currently stands. Therefore, we invite you to submit a revised version of the manuscript that addresses the points raised during the review process.

I deeply regret the unusual amount of time that has been necessary to ensure the appropriate revisions of this manuscript. While the study is intriguing, some revisions are necessary for clarity and robustness. Please enhance descriptions of research methodology, ensuring transparent reporting of allocation concealment and specifying the currency for financial figures. Clarify key elements in the statistical analysis, including outcome variable details and the rationale behind significance level choices. Ensure consistency and coherence in presenting figures, considering their placement in tables and addressing discrepancies in resilience change reporting. Additionally, respond to reviewer queries on the UCLA Loneliness scale, coding decisions for missing data, and justification for statistical significance thresholds. Attend to formatting errors and typos throughout the manuscript. Lastly, provide essential information, such as ethics committee protocol details, ISRCTN registry number, strategies for optimizing response rates, and insights into minimizing technical difficulties. I look forward to reviewing the updated version.

We look forward to receiving your revised manuscript.

Kind regards,

César González-Blanch, PhD

Academic Editor

PLOS ONE

Journal Requirements:

‘Financial support for this RCT was provided by the United States Agency for International Development under the Cooperative Agreement No. AID-OAA-A-15-00046

https://www.usaid.gov

Reviewers' comments:

Reviewer's Responses to Questions

**Comments to the Author**

1. Is the manuscript technically sound, and do the data support the conclusions?

Reviewer #1: Partly

Reviewer #2: Yes

Reviewer #3: Yes

Reviewer #4: Yes

2. Has the statistical analysis been performed appropriately and rigorously? 

Reviewer #1: No

Reviewer #2: Yes

Reviewer #3: Yes

Reviewer #4: I Don't Know

3. Have the authors made all data underlying the findings in their manuscript fully available?

Reviewer #1: Yes

Reviewer #2: Yes

Reviewer #3: Yes

Reviewer #4: No

4. Is the manuscript presented in an intelligible fashion and written in standard English?

Reviewer #1: Yes

Reviewer #2: Yes

Reviewer #3: Yes

Reviewer #4: Yes

5. Review Comments to the Author

Reviewer #1: The study is quite interesting where it investigates whether the interactive chatbot, Vitalk, is more effective than the passive use of internet resources in improving mental wellbeing and resilience outcomes of health workers in Malawi.

However, the manuscript could be further improved.

Comments

Line 115, Line 206, the description of allocation concealment is to be mentioned.

Line 165, Line 168, for $10, the currency involved e,g in USD is to be clearly stated.

Line 167, for ‘web resources At the startup’ small a for At

Line 195, the outcome variable, 1 or 2-tailed test used in the sample size calculation, what was the ‘difference’ refers to is to be stated.

Line 365, the sentence ‘days every week; and I did not work at all (coded as missing)’ not clear.

Information on missing data (if any) e.g. percentage, and type of missing data is to be stated.

Line 381, difference-in-differences (DiD) estimator is to be defined.

Line 413, p ≤ 0.05 is preferable to p≤ 0.10 as indicative of statistical significance. Consideration of low P values (e.g., P<0.10) as trending toward statistical significance may be clinically relevant for improving practice, particularly in smaller studies.

Line 450, to state around 29.

Line 454, to state around 8.

Line 458, what ‘except Age’ refers to not clear.

Line 462, comma for figures to be replaced with dot

Line 462-472, figures to be presented in table.

Line 481, typo 11.4 (should be 11.04)

Line 462-472, median and range values are to be omitted since mean ± sd is used.

Line 554, resilience change is not the same for both groups.

Line 594, the statement is incorrect.

Line 599-600, Line 636 -637, Line 639-640, the data to be presented in table form.

Line 612, surgery (depression -1.55 points)

Line 613, special care (depression -2.05)

Table 4, median score to be omitted. Frequency to be replaced with ‘n’

Line 526, the sentence is unclear.

Line 543, either mean or median scores are to be reported.

The percentage figures in text are to be reported at least 1 decimal point.

Table 6, statistical test to be denoted in the table footnote.

Line 602, 659, symbol <= to be replaced with symbol ≤

Line 656, Line 659 the significant level 0.05 to be used.

Ensure all figures cited in the text are to be cited exactly from table(s).

95%CI figures presented in the text could be placed in the table instead.

Reviewer #2: Thank you for the paper. I found the study very interesting.

The need for such as tool is extremely high in Malawi given the very limited access to psychiatric care.. Evaluation of the value of chatbots in low-resource settings and specifically for helping health worker mental health is extremely important. I am pleased to see such a trial having been conducted and even more pleased that the results show the effectiveness of such an intervention.

The recruitment of over 1500 health workers and the retention rates are great achievements. The study has been well-designed, analysed and written up. The study conducted is well-designed and gathers data over an extended period (8 weeks) with mid-study data collection. A good number of participants were included.

Regardless of the intervention and its efficacy, the results that analyse, for example depression, anxiety and burnout, for different cadres is interesting information that may not be available otherwise.

Limitations are acknowledged.

Questions and further specific comments appear below.

It is great that the study provided funding to provide data package and transportation costs. I would hope that Vitalk could be deployed without disadvantage or bias due to cost of data packages. How much technical or other support was needed for participants to use the tool on their phones? I would hope that financial, technical and other support could be provided in the future from other sources to ensure the tool is able to be useful.

Do the authors believe the tool will continue to be useful in this COVID-normal and post-COVID period? What challenges and possible solutions are envisaged for the future maintenance, scalability, deployment and ongoing provision of Vitalk?

It was good to hear that there was a risk management process in terms of recruitment exclusion criteria and triggering via PHQ-9 responses. Also clarifying the chatbot does not seek to replace a human expert and help should be sought if needed.

I think removal of NLU is a wise move to avoid errors which may occur and increase the risk of harmful or inppropriate chatbot responses.

Viki is in Figure 3 not in Figure 2.

What was the motivation for including the UCLA Lonliness scale?

Why was 'I did not work at all' coded as missing?

The sample size used significance level of .05. Why was P ≤ 0.10 used for statistical significance? This is unusual. Most commonly p<=.05 is used.

Do you have any thoughts on the differences in numbers between the two groups at different study periods. The addition of percentages of the total might be useful to aid the comparisons. Why did the control group have fewer participants at midline and post-endline and more at baseline and endline?

Only participant who completed 

Only participants who completed

I am surprised that almost all participants did not report that the pandemic had disrupted their work.

p. 23 Why wasn't mood status for the control group measured?

Control group is lower on baselines, what is the possible impact on interpretation of other results of these sig diffs. This means it would have been harder to show a difference following treatment.

So there is a significant difference for control as well, but no significant difference between groups at endline. This does not support that the intervention was useful.

Resilence is the exception.

p.24 deceases in depression 

decreases in depression

p. 25 printing /formatting error at line 504.

Only DiD estimators seem to show statistically significant differences. Why is this the case, when t-tests do not show significant differences?

p.30 "None of the 95% CI contain zero, indicating that both study groups experienced a positive effect"

How do you account for this? Were the resources potentially useful, though they were not accessed very much. Or is being alerted to how you are feeling, via doing the baseline enough to help you feel better or feel that someone else cares to ask?

p.31 "very good (greater and 0.80)"

 "very good (greater than 0.80)"

p. 31 "did not impart whether these were of clinical relevance."

impart??

Reviewer #3: This is an interesting paper reporting on the first RCT of a mental health app for healthcare workers during the COVID19 pandemic in Southern Africa combining multiple mental wellbeing outcomes, and measuring resilience and resilience-building activities. It is a well formulated and reported study which is of relevance to the mental health literature relating to the COVID-19 pandemic.

Reviewer #4: The writing is clear and straightforward. The author presents the material in an organized fashion, and the figures are well-chosen. However, I have some comments.

1. The University Research Co. (URC) Institutional Review Board and the University of Malawi Research Ethics Committee (UNIMAREC) approved the study protocol – Please kindly provide the protocol number given by ethics committee.

2. Trial was registered retrospectively with the ISRCTN registry-Please kindly provide this number.

3. How was missing data handled?

4. Strategies used to optimize the response rate were not mentioned.

5. The method of questionnaire administration was not specified.

6. There was no additional details regarding an incentive for questionnaire completion provided?

7. Response rate was not defined and also it was not reported. What was done with incomplete surveys if any?

8. Demographic data of the survey respondents was not clear. Though 8 categories of health care professionals was mentioned yet the figure only shows nurses.

9. Were the questionnaires pilot tested?

10. How can technical difficulties minimized for others who plan similar studies as there was a high attrition rate in the Treatment group due to this?

11. All the questionnaires used should be provided as appendices.

6. PLOS authors have the option to publish the peer review history of their article (what does this mean?). If published, this will include your full peer review and any attached files.

Reviewer #1: No

Reviewer #2: No

Reviewer #3: **Yes: **Dr Sarah Markham

Reviewer #4: **Yes: **Khizra Sultana

---

## [Author Response · Author response to Decision Letter 0]

9 Mar 2024

Dear editorial reviewers,

Thank you for these detailed and constructive reviews. We believe that all the changes have substantially improved the submission. The research team has addressed all the PLOS ON reviewer comments and questions in the revised manuscript. The revised manuscript and unedited manuscript have been uploaded, and responses to revisions are detailed below (in italics). We hope this clarifies all the comments and feedback. We would be happy to address any remaining questions.

Journal Requirements:

We reviewed the guidelines and made adjustments to the manuscript.

‘Financial support for this RCT was provided by the United States Agency for International Development under the Cooperative Agreement No. AID-OAA-A-15-00046

https://www.usaid.gov

We added the funding statement to the revised cover letter.

We have updated our data availability statement in the cover letter and submitted the data as supplemental information (S2).

Reviewers' comments:

Comments to the Author

1. Is the manuscript technically sound, and do the data support the conclusions?

Reviewer #1: Partly

Reviewer #2: Yes

Reviewer #3: Yes

Reviewer #4: Yes

We have reviewed our manuscript and are confident that it meets the criteria as described above. We have clearly described the methodology, the primary and secondary outcomes, and based our conclusions on the findings.

2. Has the statistical analysis been performed appropriately and rigorously?

Reviewer #1: No

Reviewer #2: Yes

Reviewer #3: Yes

Reviewer #4: I Don't Know

We have used three commonly accepted analytic methods for this type of RCT involving mental health outcomes using validated scales: mixed-effects linear models to assess difference-in-differences estimators, effect size estimates using Cohen’s d, and reliable change in risk levels. We would side with reviewers 2 and 3 that our analysis is appropriate and rigorous.

3. Have the authors made all data underlying the findings in their manuscript fully available?

Reviewer #1: Yes

Reviewer #2: Yes

Reviewer #3: Yes

Reviewer #4: No

A minimal anonymized data set necessary to replicate the study finding is uploaded as Supporting Information.

4. Is the manuscript presented in an intelligible fashion and written in standard English?

Reviewer #1: Yes

Reviewer #2: Yes

Reviewer #3: Yes

Reviewer #4: Yes

We copy edited the entire manuscript and made the appropriate changes as markups.

5. Review Comments to the Author

Reviewer #1: The study is quite interesting where it investigates whether the interactive chatbot, Vitalk, is more effective than the passive use of internet resources in improving mental wellbeing and resilience outcomes of health workers in Malawi.

However, the manuscript could be further improved.

Comments

The line numbers changed from the original due to formatting changes to comply with PLOS ONE style guidelines and additional edits. Line numbers in our responses refer to the revised manuscript with all markups shown.

Line 115, Line 206, the description of allocation concealment is to be mentioned.

Added language to lines 119-121 and 217-221

Line 165, Line 168, for $10, the currency involved e,g in USD is to be clearly stated.

Clarified on lines 171 and 174.

Line 167, for ‘web resources At the startup’ small a for At

Corrected on line 173, a period was missing.

Line 195, the outcome variable, 1 or 2-tailed test used in the sample size calculation, what was the ‘difference’ refers to is to be stated.

Clarified on lines 201-202

Line 365, the sentence ‘days every week; and I did not work at all (coded as missing)’ not clear.

Clarified on lines 380-381

Information on missing data (if any) e.g. percentage, and type of missing data is to be stated.

Thank you for pointing this out, this is important. We addressed missing data on lines 463-468, added numbers and percentages to Table 3, and added the following text. Using complete case analysis (CCA), only participants who finished base- and endline assessments were included in the calculations of DiD estimators using mixed-effects linear modeling, effect size estimation, and change in risk levels. We used CCA because missing data seem to be random as shown in the CONSORT diagram (Fig 1), which includes access to the Vitalk app or website issues, and conditionally independent of the study outcome measures. Moreover, study population characteristics of all participants with valid mental health and resilience assessments were very similar in the control and treatment groups (Fig 4). Table 3 shows the effective sample size and missing data by type of assessment. The much higher proportion of missing data for the loneliness and resilience building assessments is because participants had difficulties accessing these assessments, which were not part of the original Vitalk app but were added for this study.

Line 381, difference-in-differences (DiD) estimator is to be defined.

Added to lines 400-403. The DID estimator compares the differences in outcomes before and after treatment for the treatment group with any changes in the control group. It is a measure of additional change between base- and endline in the treatment group, if the value is statistically significant at a 0.05 level or smaller.

Line 413, p ≤ 0.05 is preferable to p≤ 0.10 as indicative of statistical significance. Consideration of low P values (e.g., P<0.10) as trending toward statistical significance may be clinically relevant for improving practice, particularly in smaller studies.

We agree and added similar language on lines 435-437.

Line 450, to state around 29.

Done on line 488

Line 454, to state around 8.

Done on line 492

Line 458, what ‘except Age’ refers to not clear.

Clarified as follows on lines 496: (n = 836 [481 control group, 355 treatment group] for all variables except Age with n = 828)

Line 462, comma for figures to be replaced with dot

We understand that there are two standards and would like to keep the comma separator. In scientific literature, including PLOS One it is common for numbers greater than 1,000 to use commas to separate groups of three digits.

Line 462-472, figures to be presented in table.

Done on lines 500-519, and the text was shortened accordingly to avoid repetition.

Line 481, typo 11.4 (should be 11.04)

Numbers were deleted from the text.

Line 462-472, median and range values are to be omitted since mean ± sd is used.

Done in Table 5.

Line 554, resilience change is not the same for both groups.

We checked the data again. Yes, the total change in resilience scores is greater in the treatment group (fig 9), but the reliable change with 13% was the same in the treatment and control groups, which is shown in Fig 8. Refers now to line 607.

Line 594, the statement is incorrect.

We added language about non-significant results at the 0.05 level on line 652.

Line 599-600, Line 636 -637, Line 639-640, the data to be presented in table form.

All data cited are presented in Table 7.

Line 612, surgery (depression -1.53 points)

This is addressed on line 668.

Line 613, special care (depression -2.06)

This is addressed on line 668.

Table 4, median score to be omitted. Frequency to be replaced with ‘n’

Done, now table 5

Line 526, the sentence is unclear.

We clarified this on lines 581-582.

Line 543, either mean or median scores are to be reported.

We are now reporting only mean scores.

The percentage figures in text are to be reported at least 1 decimal point.

We appreciate that there are different views on the number of decimals to use with percentages. We would like to stick with the NCES Statistical Standards, which states that percentages appearing in text must be rounded to whole numbers unless small differences require finer breakdowns. Using decimals in our case would suggest a level of precision and clinical significance that the self-assessments we used in our trial and our data do not justify.

Table 6, statistical test to be denoted in the table footnote.

Cited on line 657/658.

Line 602, 659, symbol <= to be replaced with symbol ≤

Done throughout the manuscript.

Line 656, Line 659 the significant level 0.05 to be used.

We addressed this on lines 652/709-710.

Ensure all figures cited in the text are to be cited exactly from table(s).

We rechecked the entire manuscript to ensure that this is the case.

95%CI figures presented in the text could be placed in the table instead.

Good suggestion, and we replaced SEs with 95% CIs..

Reviewer #2: Thank you for the paper. I found the study very interesting.

The need for such as tool is extremely high in Malawi given the very limited access to psychiatric care.. Evaluation of the value of chatbots in low-resource settings and specifically for helping health worker mental health is extremely important. I am pleased to see such a trial having been conducted and even more pleased that the results show the effectiveness of such an intervention.

Thank you for the positive feedback, it is much appreciated by all authors.

The recruitment of over 1500 health workers and the retention rates are great achievements. The study has been well-designed, analysed and written up. The study conducted is well-designed and gathers data over an extended period (8 weeks) with mid-study data collection. A good number of participants were included.

Thank you, this is greatly appreciated.

Regardless of the intervention and its efficacy, the results that analyse, for example depression, anxiety and burnout, for different cadres is interesting information that may not be available otherwise.

Limitations are acknowledged.

Questions and further specific comments appear below.

It is great that the study provided funding to provide data package and transportation costs. I would hope that Vitalk could be deployed without disadvantage or bias due to cost of data packages. How much technical or other support was needed for participants to use the tool on their phones? I would hope that financial, technical and other support could be provided in the future from other sources to ensure the tool is able to be useful.

Thank you for sharing these important points. Today’s costs of accessing the internet through mobile phones are a major barrier to using an app such as Vitalk in LMICs. However, given the rapid growth of mobile networks in low-income countries over the past decade, we anticipate that costs will come down in the foreseeable future and access will be easier without resorting to subsidies. We address some of these issues in the discussion, limitations, and conclusion sections of the revised manuscript.

Do the authors believe the tool will continue to be useful in this COVID-normal and post-COVID period? What challenges and possible solutions are envisaged for the future maintenance, scalability, deployment and ongoing provision of Vitalk?

We added language on lines 958-955.

It was good to hear that there was a risk management process in terms of recruitment exclusion criteria and triggering via PHQ-9 responses. Also clarifying the chatbot does not seek to replace a human expert and help should be sought if needed.

It is good to agree.

I think removal of NLU is a wise move to avoid errors which may occur and increase the risk of harmful or inappropriate chatbot responses.

An interesting observation. We will have to face the reality that NLU will rapidly become an integral part of chatbots; something that should be carefully evaluated.

Viki is in Figure 3 not in Figure 2.

Corrected.

What was the motivation for including the UCLA Loneliness scale?

Thank you for this question. Workplace loneliness was not addressed in the i-CBT or chatbot literature, but we included it based on CBT theory. A recent meta-analysis that was not available at the time of the study showed the link between loneliness and emotional wellbeing. Lines 797-799 address this issue, and we added a new citation to the article by Bryan et al. 

Why was 'I did not work at all' coded as missing?

The questions were posed to health workers who were active in their profession for at least 12 months prior to the study. None of the work-related questions about mental wellbeing would apply to someone who did not work during the period in question.

The sample size used significance level of .05. Why was P ≤ 0.10 used for statistical significance? This is unusual. Most commonly p<=.05 is used.

We agree and changed the text accordingly on lines 435-437.

Do you have any thoughts on the differences in numbers between the two groups at different study periods. The addition of percentages of the total might be useful to aid the comparisons. Why did the control group have fewer participants at midline and post-endline and more at baseline and endline?

We added percentages to Table 2 and added an explanation related to the passive and interactive nature in the control and treatment groups respectively on lines 459-462.

Only participant who completed 

Only participants who completed

Corrected.

I am surprised that almost all participants did not report that the pandemic had disrupted their work.

Anecdotal evidence suggests that health services operated during the pandemic in Malawi.

p. 23 Why wasn't mood status for the co

---

## [Editor Report · Decision Letter 1]

25 Apr 2024

Effectiveness of a chatbot in improving the mental wellbeing of health workers in Malawi during the COVID-19 pandemic: A randomized, controlled trial

PONE-D-23-02103R1

Dear Dr. Kleinau,

We’re pleased to inform you that your manuscript has been judged scientifically suitable for publication and will be formally accepted for publication once it meets all outstanding technical requirements.

Kind regards,

César González-Blanch, PhD

Academic Editor

PLOS ONE

Additional Editor Comments (optional):

I sincerely appreciate the effort you've put into addressing the concerns and suggestions raised by the reviewers. The feedback provided by the reviewers underscores the importance and relevance of the study, as well as the robustness of its design and analysis. Significant revisions have been made to the manuscript to address the raised concerns, including improvements in data presentation, clarity of writing, and justification of methodological decisions. The detailed response to the comments demonstrates a clear commitment to the quality and rigor of the research.
---

## [Editor Report · Acceptance letter]

7 May 2024

PONE-D-23-02103R1 

PLOS ONE

Dear Dr. Kleinau, 

I'm pleased to inform you that your manuscript has been deemed suitable for publication in PLOS ONE. Congratulations! Your manuscript is now being handed over to our production team.

Kind regards, 

on behalf of

Dr. César González-Blanch 

Academic Editor

PLOS ONE